# Episodic events are flexibly encoded in both integrated and separated neural representations

Zhenghao Liu ◉ ✉, Mikael Johansson ◉ & Inês Bramão ◉ ✉

Remembering everyday events involves noticing what different experiences share and preserving the details that set them apart, yet the neural processes supporting this balance remain unclear. Here, we record EEG while participants view naturalistic movie scenes that introduce episodic events with overlapping elements. Using time-resolved representational similarity analysis, we find that these events evoke both similarities and dissimilarities in neural patterns as new information unfolds. Similarities predict successful inference of information across separate episodes, consistent with integrative encoding. Dissimilarities, by contrast, predict accurate memory for individual events, indicating the formation of distinct event-specific traces. Together, these findings indicate that the brain encodes both integrated and separated neural representations to flexibly support different mnemonic goals and to balance relational inference with detailed recollection.

Episodic memory enables the adaptive retrieval of commonalities and unique details across overlapping life events. For example, encountering a woman in the park with your colleague's daughter may lead to the formation of an integrated memory representation connecting the woman and your colleague. At the same time, you may retain a distinct representation of this specific encounter to later discuss with your colleague the circumstances of meeting her daughter. This study investigates how the brain simultaneously supports both the integration across related events and the preservation of event-specific details.

Previous research has examined how the brain represents memories for related events by analyzing the neural pattern similarities associated with different events. Some memory models suggest that the hippocampus encodes related events into distinct memory representations to minimize interference[1–3]. However, more recent evidence indicates that the hippocampus also supports the integration of related events[4,5], facilitating the inference of indirect associations between elements encountered in separated episodes[6,7]. To reconcile these apparently conflicting findings, it has been proposed that hippocampal function varies along its axis, with the anterior hippocampus supporting the formation of integrated memory representations and the posterior hippocampus forming distinct, event-specific memory representations[8,9].

Despite evidence for the formation of both integrated and separated memory representations, it remains unclear whether these representations can coexist for the same event or if integration compromises event-specific detail. Some findings suggest that memory integration comes at the cost of diminished episodic detail. In particular, Carpenter and colleagues[10,11] demonstrated that integrated representations are associated with reduced memory for episodic details, indicating a trade-off between memory integration and event-specific detail. This raises the possibility that the brain cannot simultaneously keep both types of representations. Nevertheless, this trade-off has not been consistently replicated, and some studies even report a positive association between memory integration and episodic detail memory[12–14], implying that integrated and separated representations may coexist[8,15].

It is conceivable that the brain maintains both types of representations to flexibly support distinct memory functions[9]. Integrated representations capture generalized information, facilitate inferences across events, and allow the seamless retrieval of overlapping memories. In contrast, separated representations preserve event-specific features, minimize interference, and enable the accurate retrieval of unique event details. Previous fMRI studies have shown that these two types of representations emerge in distinct hippocampal subregions[9].

Department of Psychology, Lund University, Lund, Sweden. ✉e-mail: zhenghao.liu@psy.lu.se; ines.bramao@psy.lu.se

However, it remains unclear how these representations are formed during real-time encoding, particularly when new episodic events overlap with prior experiences. Specifically, little is known about how the brain simultaneously integrates novel experiences into existing memory networks while maintaining the uniqueness of individual events. Do integration and separation processes interact dynamically during encoding, and how does the brain determine which aspects of an experience should be merged and which should remain distinct?

To address this gap, we employed time-resolved representational similarity analysis (RSA) of electroencephalographic (EEG) recordings to track the dynamic formation of neural patterns during real-time encoding of related events. This approach allowed us to examine the emergence of integrated and separated representations as they develop in real-time and to investigate how they support different memory functions. Furthermore, by examining how shared and novel information across related events influences the formation of these neural representations, we provide insight into the mechanisms that govern the balance between memory integration and separation during episodic encoding.

Neural activity, particularly in the theta (4 to 7 Hz) and alpha-beta (8 to 20 Hz) frequency bands, has been consistently implicated in the processes underlying memory formation and retrieval[16–18]. Empirical evidence demonstrates that increases in theta power often accompany successful memory performance, suggesting that these oscillations facilitate the binding of disparate elements of an experience into a coherent memory representation[16,19,20]. Recent work employing RSA has further indicated that theta-band activity supports the integration of overlapping memories, allowing related experiences to be linked[19,21]. In contrast, alpha-beta oscillations exhibit an inverse relationship with memory encoding and retrieval: decreases in alpha-beta power have been associated with improved memory performance[18,22], potentially reflecting the engagement of cortical networks necessary for active information processing and representation. Conversely, increases in alpha-beta power have been linked to inhibitory control mechanisms[23], suggesting that alpha-beta may play a role in suppressing competing or overlapping memory traces during processes that require memory separation.

We employed a naturalistic paradigm, adapted from the classic associative inference memory paradigm[5,24,25], in which participants encode videos that simulate real-life events, allowing us to capture the complexity of memory formation in an ecologically valid setting. Participants encoded movies featuring real-world interactions between Sim characters (https://www.ea.com/games/the-sims/the-sims-4). First, they encoded AB movies, in which a Sim A interacted with a Sim B. Later, they encoded BC movies, where a new Sim C interacted with the previously seen Sim B. A control condition featured interactions between two novel Sims (XY movies). During retrieval, participants completed memory tests for both direct (i.e., AB, BC, XY) and indirect (i.e., AC) associations. These tests were interleaved, with the constraint that AC associations were tested before their corresponding AB and BC associations. Each association memory test was followed by a source memory task, where participants indicated if the two characters had directly interacted. Participants also rated their confidence (see Fig. 1A). High AC performance was considered an indicator of the formation of integrated representations across events[5,24,25], while high source memory performance reflected preserved event-specific representations (see also[10,26]). An additional surprise memory test for episodic details (context and character clothing) showed no systematic effects of AC retrieval and is reported in the supplementary material (see Supplementary Note 1).

EEG data were recorded continuously throughout the experiment. We applied RSA to the EEG signals during the encoding of AB, BC and XY movies, allowing us to track the formation of both integrated and separated memory representations (see Fig. 2). Memory integration was expected to manifest as greater neural similarity between AB and BC movies, whereas memory separation was anticipated to be reflected in increased neural dissimilarity between AB and BC movies relative to control conditions. In addition to RSA, we conducted a univariate time-frequency analysis to examine the relationship between oscillatory activity and observed neural similarities and

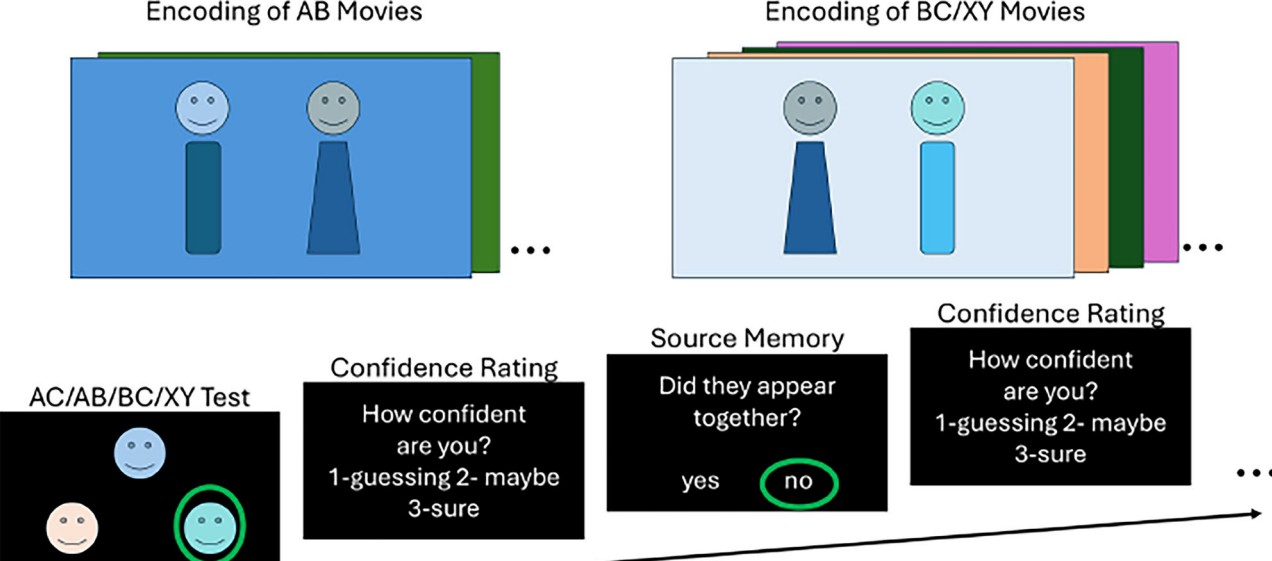

**Fig. 1 | Overview of the experimental paradigm.** Participants first encoded AB movies in which they encountered two characters. Later, they encoded BC movies, where a new character interacted with a previously seen one. XY movies were also presented with two completely new characters and serve as a control condition. Each movie was presented five times during encoding. At retrieval, participants completed memory tests for direct (AB, BC and XY) and indirect associations (AC). The retrieval test order was randomized, with the constraint that AC associations were always tested before their corresponding AB and BC associations. Following each memory test, participants completed a source memory test to indicate if the two characters had been seen together. Correct responses are identified with green circles for illustration only, these visual markers were not shown to participants during the actual task. Confidence ratings were also registered. The experimental stimuli were created using The Sims 4 (© Electronic Arts Inc.). The images displayed are placeholders provided solely for illustrative purposes.

**Fig. 2 | Analytical pipeline and predictions of the present study.** Experimental stimuli were created using The Sims 4 (© Electronic Arts Inc.). Placeholder figures are presented here for illustration. **A** Trial structure. Each movie began with the display of Sim A/C/Y for two seconds, followed by a three-second animation of this Sim in context. Afterward, a one-second fixation cross was presented, followed by the presentation of the Sim B/X for two seconds. The movie ended with a five-second animation of the two Sims AB/BC/XY interacting in a context. We measured the neural representational similarity of the 'Sim A and B in Context' of AB movie (outlined in red) and each timepoint of the corresponding BC movie (outlined in red). **B** Feature Selection. To identify the time-frequency features that were sensitive to the content of the AB movies, we compared the wavelet coherence between different repetitions of the same AB movie against the coherence between different AB movies. **C** A time-resolved neural representational similarity analysis was performed by correlating the representative features of the AB movie with the patterns of the corresponding BC movie across the entire encoding period. Systematic similarities and dissimilarities were estimated by contrasting the neural patterns similarities between AB and corresponding BC movie against two baselines: 1) neural similarities between AB and XY movies, and 2) neural similarities between AB and non-corresponding BC movies. **D** Systematic similarities, indicative of memory integration, should predict AC retrieval performance, whereas systematic dissimilarities, indicative of memory separation, should predict source memory performance.

dissimilarities. We predicted that neural similarities would be associated with increased theta power and decreased alpha/beta power, whereas neural dissimilarities would correspond to increased alpha/beta activity.

Furthermore, we hypothesized that neural similarities would predict AC retrieval, while neural dissimilarities would predict source memory performance. Specifically, if integrated and separated

representations coexist, successful AC retrieval should be accompanied by better source memory, indicating that both types of representations, supporting different memory functions, were encoded for the same events. Conversely, if integration disrupts event-specific details, successful AC retrieval should impair source memory, and the neural patterns predictive of AC retrieval would associate with poorer source memory performance.

## A. Performance for AC Inference and Direct Association Memory

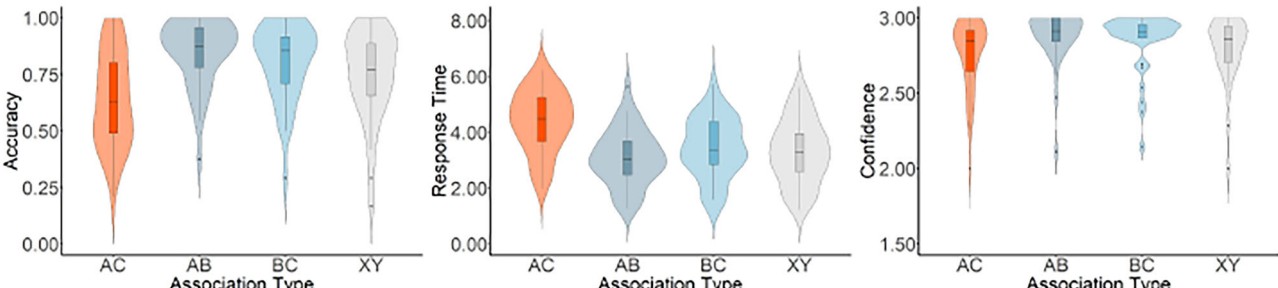

## B. Performance for Source Memory

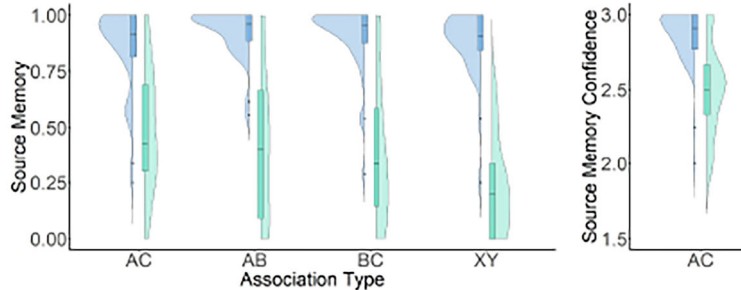
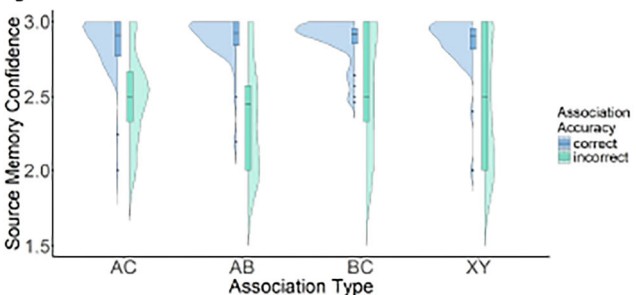

**Fig. 3 | Behavioral result summary. A** Accuracy, response time, and confidence for the memory tests (AC vs AB vs BC vs XY). Violin plots show the sample distribution, with box plots overlaid. The central line of the box plot indicates the mean, the box represents the interquartile range (25th-75th percentile), and whiskers extend to 1.5 inter quantile range. Individual data points beyond the whiskers are plotted as dots. Plots were generated using data from 36 participants in a repeated-measures design. **B** Accuracy and confidence for the source memory tests of each association type (AC vs AB vs BC vs XY) and retrieval accuracy (Correct vs. Incorrect) of the corresponding association test. Raincloud plots show the distribution of the data, with accompanying box plots indicate summary statistics. The mean is shown by the central bar, and the box captures the middle quartiles (25th-75th percentile). Whiskers reflect 1.5 interquartile range, and any values outside this span are displayed as dots.

## Results

### Associative memory tests

Memory performance for direct and indirect associations is summarized in Fig. 3A. We contrasted memory performance across the four association types (AC vs. AB vs. BC vs. XY) with linear mixed-effects models. This analysis showed significant main effects of association type on accuracy ($F(3, 3420) = 48.068$, $p < 0.001$, $\eta_p^2 = 0.04$, 95% CI = [0.03, 1.00]), response time ($F(3, 2587) = 71.314$, $p < 0.001$, $\eta_p^2 = 0.08$, 95% CI = [0.06, 1.00]) and confidence ($F(3, 2584) = 11.921$, $p < 0.001$, $\eta_p^2 = 0.01$, 95% CI = [0.01, 1.00]). Post-hoc Tukey-corrected comparisons showed that retrieval of AC associations was associated with significantly lower accuracy, slower response times, and reduced confidence compared with all the direct associations, except that the lower confidence of AC compared to XY did not reach statistical significance (accuracy − AB: $t(3423) = -11.29$, $p < 0.001$, $D = -0.212$, 95% CI = [-0.318, -0.105]; BC: $t(3423) = -9.07$, $p < 0.001$, $D = -0.170$, 95% CI = [-0.256, -0.085]; XY: $t(3423) = -5.92$, $p < 0.001$, $D = -0.111$, 95% CI = [-0.167, -0.055]; response time − AB: $t(2590) = 13.75$, $p < 0.001$, $D = 0.456$, 95% CI = [0.226, 0.685]; BC: $t(2590) = 9.12$, $p < 0.001$, $D = 0.305$, 95% CI = [0.152, 0.459]; XY: $t(2590) = 11.77$, $p < 0.001$, $D = 0.401$, 95% CI = [0.199, 0.603]; confidence − AB: $t(2590) = -5.47$, $p < 0.001$, $D = -0.099$, 95% CI = [-0.149, -0.049]; BC: $t(2590) = -4.66$, $p < 0.001$, $D = -0.085$, 95% CI = [-0.128, -0.042]; XY: $t(2590) = -2.43$, $p = 0.072$, $D = -0.045$, 95% CI = [-0.094, 0.004]).

Furthermore, XY associations were retrieved less accurately than both AB ($t(3423) = 5.37$, $p < 0.001$, $D = 0.101$, 95% CI = [0.041, 0.161]) and BC ($t(3423) = 3.15$, $p = 0.009$, $D = 0.059$, 95% CI = [0.015, 0.103]) associations, while no statistical difference was found between the AB and BC retrieval accuracy ($t(3423) = 2.22$, $p = 0.118$, $D = 0.042$, 95% CI = [-0.011, 0.095]). In terms of response times, BC was retrieved slower than both AB ($t(2590) = -4.85$, $p < 0.001$, $D = -0.150$, 95% CI = [-0.173, -0.127]) and XY ($t(2590) = -2.98$, $p = 0.015$, $D = -0.096$, 95% CI = [-0.174, -0.018]), likely reflecting proactive interference due to the shared *Sim B* (see references[27,28] for a similar effect). No significant differences were observed between AB and XY associations ($t(2591) = -1.73$, $p = 0.310$, $D = -0.054$, 95% CI = [-0.160, 0.052]). Finally, AB associations were retrieved more confidently than XY associations ($t(2591) = 3.11$, $p = 0.010$, $D = 0.054$, 95% CI = [0.014, 0.094]), while the confidence of BC was not significantly different from either AB ($t(2590) = 0.19$, $p = 0.848$, $D = 0.014$, 95% CI = [-0.127, 0.155]) or XY retrieval ($t(2590) = 1.64$, $p = 0.101$, $D = 0.040$, 95% CI = [-0.008, 0.088]).

### Source memory test

To evaluate how source memory performance varied as a function of retrieval accuracy across the different associations, we used linear mixed-effects models with Association Type (AC vs. AB vs. BC vs. XY) and Retrieval Accuracy (Correct vs. Incorrect) as factors, see Fig. 3B. The results show a significant main effect of Association Type for accuracy ($F(3, 36) = 5.329$, $p = 0.004$, $\eta_p^2 = 0.31$, 95% CI = [0.08, 1.00]) but not for confidence ($F(3, 126) = 1.947$, $p = 0.125$, $\eta_p^2 = 0.04$, 95% CI = [0.00, 1.00]). Specifically, source memory for XY was worse compared with both AC ($t(33) = -3.36$, $p = 0.010$, $D = -0.083$, 95% CI = [-0.146, -0.020]) and AB associations ($t(30) = -3.23$, $p = 0.015$, $D = -0.108$, 95% CI = [-0.177, -0.039]). All the other contrasts were non-significant (For accuracy: AC-AB: $t(29) = -0.85$, $p = 0.832$, $D = -0.025$, 95% CI = [-0.258, 0.208]; AC-BC: $t(31) = 0.23$, $p = 0.996$, $D = 0.006$, 95% CI = [-1.995, 2.007]; AB-BC: $t(23) = 1.25$, $p = 0.605$, $D = 0.031$, 95% CI = [-0.087, 0.149]; BC-XY: $t(32) = 2.44$, $p = 0.091$, $D = 0.077$, 95% CI = [-0.014, 0.168]. For confidence: AC-AB: $t(101) = 1.27$, $p = 0.582$, $D = 0.045$, 95% CI = [-0.115, 0.205]; AC-BC: $t(83) = -0.83$, $p = 0.842$, $D = -0.030$,

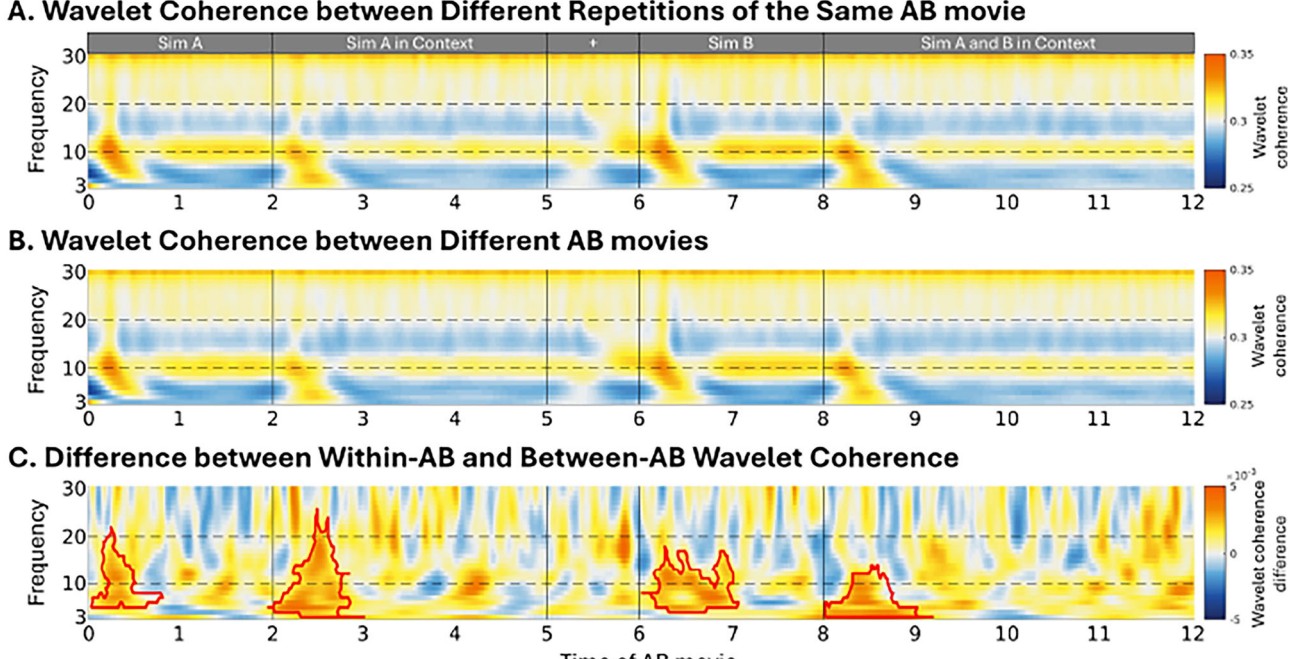

**Fig. 4 | Time–frequency feature identification. A** Within-AB wavelet coherence, **B** between-AB wavelet coherence, and (**C**) their coherence differences. Significant clusters, marked by red edges, indicate the time-frequency features that were stable across the five AB repetitions and sensitive to different AB movie content. The identified features were used as a template to extract the power values of the AB movies, which were used in the subsequent representational similarity analysis.

95% CI = [-0.321, 0.261]; AC-XY: $t(78) = 0.76$, $p = 0.873$, $D = 0.026$, 95% CI = [-0.289, 0.341]; AB-BC: $t(369) = -2.22$, $p = 0.120$, $D = -0.075$, 95% CI = [-0.169, 0.020]; AB-XY: $t(313) = -0.57$, $p = 0.942$, $D = -0.019$, 95% CI = [-0.519, 0.482]; BC-XY: $t(291) = 1.72$, $p = 0.317$, $D = 0.056$, 95% CI = [-0.054, 0.166]). The main effect of Retrieval Accuracy was also significant for both accuracy ($F(1, 26) = 293.667$, $p < 0.001$, $\eta_p^2 = 0.92$, 95% CI = [0.86, 1.00]) and confidence ($F(1, 2656) = 282.903$, $p < 0.001$, $\eta_p^2 = 0.10$, 95% CI = [0.08, 1.00]), showing that when the associations were correctly retrieved, source memory performance was higher and also associated with higher confidence ratings.

Additionally, a significant interaction between Association Type and Retrieval Accuracy was observed for both source memory accuracy ($F(3, 34) = 11.007$, $p < 0.001$, $\eta_p^2 = 0.49$, 95% CI = [0.26, 1.00]) and confidence ($F(3, 2523) = 2.802$, $p = 0.039$, $\eta_p^2 = 3.32e-3$, 95% CI = [0.00, 1.00]). Post-hoc tests confirmed that higher source memory accuracy and confidence were consistently associated with correct association retrieval (AC: $t(33) = 9.50$, $p < 0.001$, $D = 0.372$, 95% CI = [0.295, 0.449]; AB: $t(30) = 8.16$, $p < 0.001$, $D = 0.486$, 95% CI = [0.369, 0.603]; BC: $t(31) = 10.39$, $p < 0.001$, $D = 0.500$, 95% CI = [0.405, 0.595]; XY: $t(32) = 15.76$, $p < 0.001$, $D = 0.616$, 95% CI = [0.539, 0.693]), suggesting that the observed interactions were primarily driven by the effect size differences across Association Types. Monte Carlo sampling further revealed that the differences in source memory accuracy between correct and incorrect retrieved associations were largest for XY associations (Monte Carlo $p$s < 0.001), whereas the other associations showed comparable levels of this difference (Monte Carlo $p$s > 0.128). On the other hand, the effect size of retrieval accuracy on source memory confidence was smallest for BC associations (Monte Carlo $p$s < 0.050), while the effect size showed no significant differences across the other association types (Monte Carlo $p$s > 0.346).

In summary, the behavioral results show that source memory performance is positively associated with AC memory performance, supporting the idea that related events may be simultaneously encoded in both integrated and separated memory representations. These different memory representations may facilitate distinct memory functions and support different mnemonic task demands.

## Feature selection reveals time-frequency signatures of AB movie content

The EEG data collected during the encoding of each AB movie were transformed into a time-frequency representation (TFR) for use in the neural representational similarity analysis (RSA). Before conducting the RSA, we identified representative time-frequency features that reliably captured the contents of the AB movie. Those features were expected to remain stable across repetitions of the same AB movie and distinct across different AB movies. To achieve this, we applied a wavelet coherence-based approach, which is highly sensitive to subtle phase shifts in neural signals[29], such as those expected across the five repetitions of each movie[30]. Importantly, wavelet coherence can be computed across all channels, timepoints, and frequency bands, preserving the multidimensional structure of the EEG data. This approach ensured that the features selected for the RSA accurately reflected the dynamic neural representations of each movie while accounting for temporal variability across repetitions. Specifically, we computed the coherence across the five repetitions of each AB movie (within-AB coherence) and contrasted it with the coherence between the different AB movies (between-AB coherence; see Fig. 2B).

Time-frequency features exhibiting significantly higher within-AB coherence than between-AB coherence were considered sensitive to the context of AB and selected for the RSA. Figure 4 presents the results of this feature selection, averaged across all AB movies. A cluster-based permutation test revealed four significant clusters. The first cluster emerged immediately after the movie onset, between 0–0.9 sec, corresponding to the initial presentation of Sim A ($t$-cluster = 1.229e + 4, $p = 0.003$). A second cluster appeared between 2–3 sec, capturing the representation of Sim A within its context ($t$-cluster = 3.344e + 4, $p < 0.001$). The third cluster, occurring between 6-7.1 sec, reflected the distinct neural representation of Sim B ($t$-cluster = 1.821e + 4, $p = 0.001$). Finally, the fourth cluster, observed between 8–9.2 sec, corresponded to when both Sim A and B appeared within a context, capturing the full event representation ($t$-cluster = 3.251e + 4, $p < 0.001$).

These AB movie-specific features were then used as a template to extract the power values from the TFR of AB trials, which were then used to estimate their time-resolved representational similarity with the same TFR features extracted from the XY and BC trials at each time point. Our analysis focused on the similarity between the time window corresponding to the Sim A and B interaction (i.e., the segment of 'Sim A and B Context') and each timepoint of the XY and BC movies encoding. This segment was chosen as it encapsulates the complete event structure, including both Sims, their actions, and the context (See Fig. 2A). Additional analyses investigating the representational similarity between the other AB movie segments and XY and BC movies are reported in Supplementary Note 7.

Moreover, to demonstrate that the representational similarity approach employed here successfully captures similarities driven by perceptual input, we compared the neural similarity across repetitions of the same AB movie against that of different AB movies. The results showed clear neural representational similarities driven by perceptual input, providing a prerequisite for investigating memory-related neural similarities (see Supplementary Note 3).

## The neural representations of AB and BC movies share both similarities and dissimilarities

To investigate how AB and BC movies are represented at the neural level, we conducted a representational similarity analysis (RSA). First, we extracted time-frequency features sensitive to the content of each AB movie, averaging across the five repetitions to ensure robustness. To account for potential time shifts across repetitions, we used the temporal median of these features[31]. Next, for each participant and AB movie, we correlated these extracted features with the TFR of the corresponding BC movie at each time point during BC encoding, resulting in a time-resolved representational similarity matrix. This matrix captures the systematic similarities and dissimilarities between the neural representations of AB and the corresponding BC movie. The BC movies began with the segments involving only the novel Sim C, followed by the common Sim B. Thus, when participants were first exposed to the novel Sim C, they could not yet know which corresponding AB movie it was associated with. This may have introduced distinct patterns of similarities and dissimilarities in this initial viewing, which was therefore excluded from the analysis (see Supplementary Note 5 for RSA across all repetitions).

To determine whether the neural representations of AB and BC movies were systematically related, we first contrasted the representational similarity between each AB movie and its corresponding BC movie against the similarity between AB movies and all XY movies. Bayesian statistical analysis revealed systematic similarities and dissimilarities in several time windows (see Fig. 5A). Specifically, increased similarity was observed centered at 1 second (average similarity = 1.1e-2, minimal $BF_{10}$ in this time window = 3.166), during the segment of 'Sim C', followed by two additional peaks at 2.8 sec (average similarity = 9.0e-3, minimal $BF_{10}$ in this time window = 3.151) and 3.7 sec (average similarity = 1.2e-2, minimal $BF_{10}$ in this time window = 3.061), during the presentation of 'Sim C in Context'. In contrast, increased dissimilarities were observed later during BC encoding, at 5.1 seconds (average similarity = -8.9e-3, minimal $BF_{10}$ in this time window = 3.164) and 7.4 sec (average similarity = -1.3e-2, minimal $BF_{10}$ in this time window = 3.185), immediately before and during the presentation of the segment of 'Sim B'.

Next, we compared the similarities between the AB movie and its corresponding BC movie with those evoked by the AB movie and all non-corresponding BC movies. While using the XY movies is a common baseline in the literature (for a review, see ref. 8), it does not entirely rule out general, content-unspecific reinstatement. The Sim B might trigger general retrieval processes in which previously encoded events are broadly reactivated in search of the corresponding AB movie. This could influence the observed representational similarity,

regardless of whether the movies share a common element. Thus, to differentiate content-specific reinstatement (i.e., reinstatement of the corresponding AB movie) from the general reinstatement (i.e., broader reinstatement of multiple AB movies) we compared the similarity of AB and corresponding BC movie against the similarity between AB and all non-corresponding BC movies. This analysis allowed us to isolate neural patterns reflecting the retrieval of a specific BC movie rather than general memory reinstatement. The results of this analysis replicated the previous findings (see Fig. 5B), with similarities emerging at 1 sec (average similarity = 7.5e-3, minimal $BF_{10}$ in this time window = 3.019), 2.8 sec (average similarity = 8.0e-3, minimal $BF_{10}$ in this time window = 3.188) and 3.7 sec (average similarity = 9.2e-3, minimal $BF_{10}$ in this time window = 3.065) after BC movie onset. Additionally, the same dissimilarity at 7.4 sec was observed (average similarity = -8.8e-3, minimal $BF_{10}$ in this time window = 3.164), confirming that our findings are not merely the result of broader retrieval processes but instead reflected content-specific neural similarities and dissimilarities.

In Supplementary Note 4, we examine the topographical distribution of the observed similarities and dissimilarities. The analysis revealed that these effects were broadly distributed across the scalp. In Supplementary Note 5, we further report how the similarities and dissimilarities evolved across the five repetitions of BC movie encoding. Across repetitions, we observed incremental shifts in both, reflecting an adaptive process in which integrated and separated representations changed dynamically over time. Finally, to test for a potential trade-off between integration and separation, we calculated correlations between trial-level similarities and dissimilarities. The results showed that similarities tended to co-occur within the same trial but did not significantly correlate with dissimilarities. Together, at the trial level, there is no statistical evidence of a trade-off between memory integration and separation (Supplementary Note 6).

In summary, the neural representations of AB and BC movies share both similarities and dissimilarities, revealing integrated and separated patterns. The similarities occur when the novel element, Sim C, is presented, suggesting that participants integrate overlapping elements from different experiences into unified representations. In contrast, dissimilarities arise when the common cue, Sim B, is presented, indicating a neural mechanism that preserves distinctions between similar yet distinct events, likely to minimize interference. These findings highlight the brain's ability to flexibility balance integration and separation, which is fundamental for creating associative memory links between different experiences and for maintaining episodic memory specificity.

## Neural similarities relate to associative inference and neural dissimilarity relates to source memory

Next, we investigated how the identified neural similarities and dissimilarities between AB and the corresponding BC movie relate to later memory performance. For each trial, we extracted the similarity between AB and corresponding BC, with the similarity between AB and all non-corresponding BC subtracted as a baseline, within the time windows identified in the previous analysis (see Fig. 5B). Bayesian linear regression was used to compare whether these neural similarities and dissimilarities show differences between trials with correct and incorrect indirect AC association or source memory. Both models controlled for repetition effects, and the source memory model also accounted for indirect AC association performance. To ensure balanced data, we excluded participants with near-ceiling indirect AC association or source memory performance (above 90%), and few incorrect responses. Thus, 30 participants were included in the model of AC association and 28 in the model of source memory.

We hypothesized that higher similarity between AB and corresponding BC movies, reflecting integration across related events, would be associated with higher AC retrieval. Conversely, greater

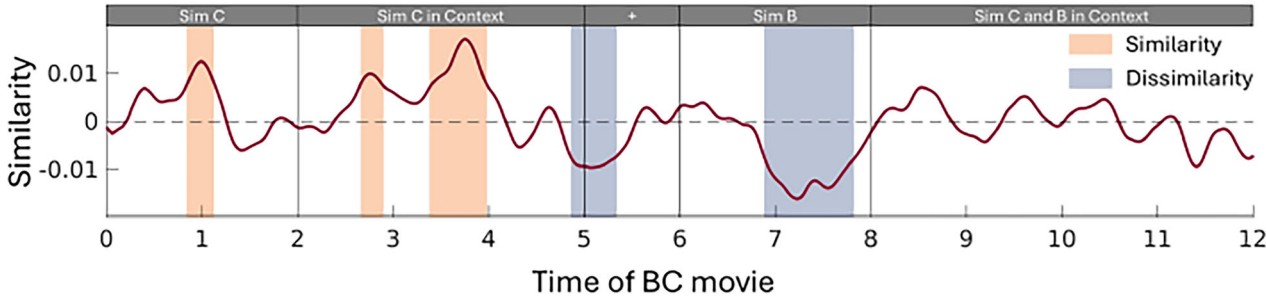

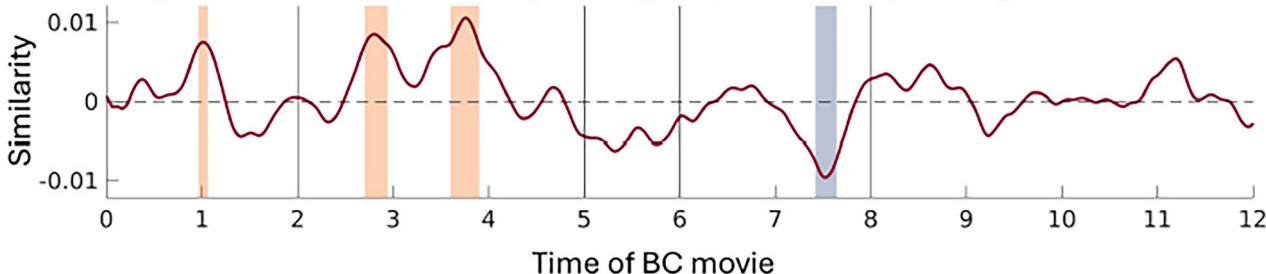

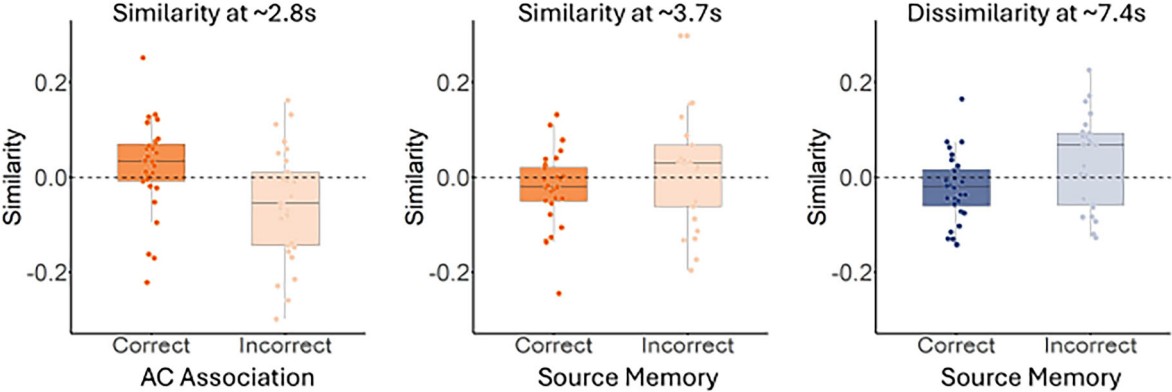

**Fig. 5 | Time-resolved representational similarities and dissimilarities between AB movie and corresponding BC movie, and their relationship with memory performances.** Time windows, identified with a Bayesian approach, are highlighted to indicate moments of increased similarity and dissimilarity. **A** Neural pattern similarities between AB and corresponding BC movie contrasted against the similarities between AB and XY movies. **B** Neural pattern similarities between AB and corresponding BC movie contrasted against the similarities between AB and non-corresponding BC movies. These findings reveal distinct phases of integration and separation in neural representations, with similarity effects emerging during Sim C encoding and dissimilarity effects occurring during the presentation of the common Sim B. **C** Representational similarities and dissimilarities for trials with correct / incorrect memories. The plotted similarities account for the effects of controlled variables and display the binary relationship between neural similarities and memory performance. The box plots visualize the distribution of (dis)similarities across participants, including 30 participants for the AC association analysis and 28 participants for the Source Memory analysis, following the exclusion of participants with ceiling performance (accuracy > 90% in the respective memory tests). Each data point reflects the average similarity for correct and incorrect memory trials per participant. The central line of the box plot indicates the mean, the box represents the interquartile range (25th-75th percentile), and whiskers extend to 1.5 inter quantile range.

dissimilarities, indicating efforts to separate related memory traces, would relate to better source memory. The observed results support our predictions. The similarity observed at 2.8 seconds was greater in trials with better AC memory performance ($\beta = 0.044$, $BF_{10} = 14.668$), suggesting that integrating events facilitates the formation of indirect associations. Additionally, the similarity observed at 3.7 seconds was negatively related to source memory performance ($\beta = -0.036$, $BF_{10} = 5.345$), indicating that integration may hinder the ability to remember specific event details. Finally, the dissimilarity observed at 7.4 seconds was associated with source memory performance ($\beta = -0.041$, $BF_{10} = 8.681$), showing that memory separation helps to preserve the source of different events (see Fig. 5C). No other effects were

statistically associated with AC or source memory performance (see Supplementary Note 8 for more details).

## Electrophysiological markers associated with the encoding of related events

A univariate analysis investigated the time-frequency patterns associated with encoding new events that share information with previously encoded experiences. A cluster-based permutation test[32] was used to contrast BC encoding against XY encoding, within each BC movie segment, and using the same frequency range (3-30 Hz) as in the previous RSA. To be consistent with the previous analysis, the initial encoding repetition of the BC and XY movies was excluded.

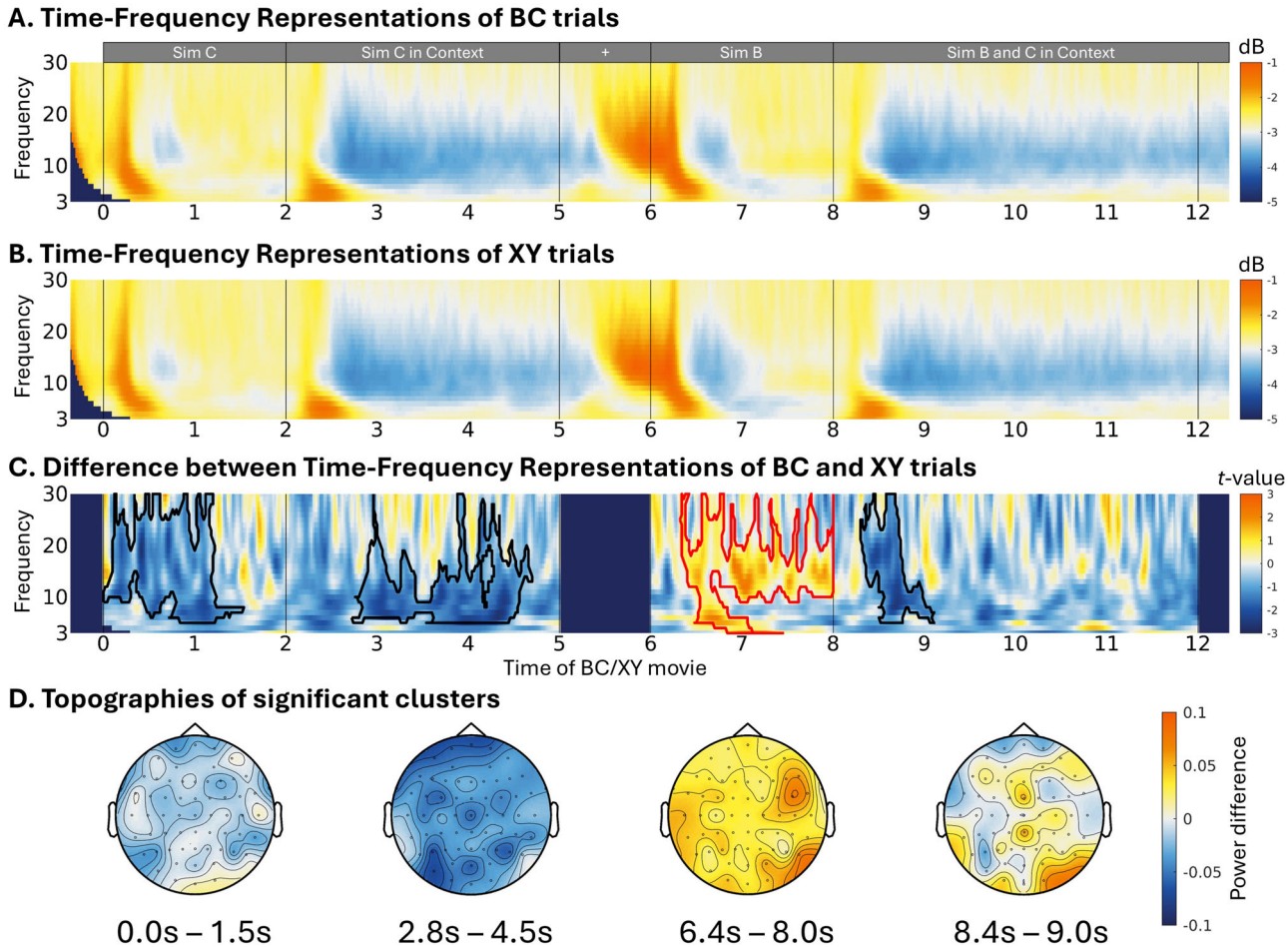

**Fig. 6 | Univariate analyses results.** Time-frequency representations of BC (**A**) and XY (**B**) encoding, covering the frequencies from 3 to 30 Hz, were transformed into dB representations for visualization purposes. A cluster-based permutation analysis was used to estimate the power differences between BC and XY encoding. The significant clusters were marked in red (for BC > XY) and black (for BC < XY) edges (**C**). The topographies of the significant clusters of differences between BC and XY are plotted in (**D**).

Our results show decreased alpha-beta power in the BC compared with the XY encoding in three different time windows: 1) between 0.0 and 1.5 sec (cluster-$t$ = -2.934e + 4, $p$ < .001), during the segment of 'Sim C'; 2) between 2.8 and 4.5 sec (cluster-$t$ = -3.202e + 4, $p$ = .009), during the segment of 'Sim C in Context', and between 8.4 and 9.0 sec (cluster-$t$ = -2.114e + 4, $p$ = .043), during the segment of the 'Sim B and C in Context'. Additionally, we observed a cluster indicating higher theta and alpha-beta power during the encoding of BC compared with XY between 6.4 and 8 sec (cluster-$t$ = 2.505e + 4, $p$ = .014), during the segment of 'Sim B' (see Fig. 6).

Interestingly, the alpha-beta power decreases roughly correspond to the time windows where neural pattern similarities were found, while the increased theta and alpha/beta were found in the time windows associated with neural pattern dissimilarities. To examine if these time-frequency representations are linked to the formation of integrated and separated memory representations, we employed Bayesian linear regression analysis to investigate if these power changes predicted the neural (dis-)similarities observed in the corresponding time windows. The analysis showed that the alpha-beta power decrease observed between 0.0 and 1.5 sec predicted the similarity centered at 1.0 sec ($\beta$ = -0.202, $BF_{10}$ = 3.754e + 29) during the segment of 'Sim C'. Furthermore, the alpha-beta power decrease found between 2.8 sec to 4.5 sec predicted the similarity centered at 2.8 seconds ($\beta$ = -0.045, $BF_{10}$ = 29.831) and ~3.7 sec ($\beta$ = −0.128, $BF_{10}$ = 5.870e + 11) during the segment of 'Sim C in Context'. Additionally, the theta and alpha-beta power increase observed between 6.4 and 8 sec predicted the

dissimilarity centered at 7.4 sec ($\beta$ = -0.107, $BF_{10}$ = 1.782e + 8) during the segment of 'Sim B'.

Previous studies have consistently reported that decreases in alpha-beta power are associated with successful memory encoding and retrieval[17,18,33,34]. Here, we provide evidence that alpha-beta power decreases are also related to the formation of integrated representations. Additionally, elevated alpha-beta power is typically linked to inhibitory control mechanisms[23]. Our findings suggest that increased alpha/beta power may be involved in establishing separated memory representations.

## Discussion

The present study investigates how memories for related events are encoded into integrated and separated representations to support various mnemonic functions. Using an adapted version of the associative inference paradigm[5,24,25] with naturalistic stimuli, Sim movies that simulate real-life interactions, we examined whether the brain flexibly encodes related events into integrated and separated memory representations. Participants first encoded a set of AB movies, followed by the encoding of a new set of BC movies, in which a familiar character interacted with a new one. A final memory test assessed participants' ability to retrieve the indirect AC associations, while a source memory test examined whether participants remembered that the two characters had not been seen together. By employing time-resolved representational similarity analysis of EEG data, we tracked similarities and dissimilarities in the neural patterns between the

encoding of the AB and BC movies. Our results show clear evidence that the human brain dynamically engages in memory integration and separation to support different aspects of memory and provide insights into the temporal dynamics of these processes.

Increased representational similarity across related events has been consistently associated with forming integrated memory representations, which facilitates the linking of information from separate episodes into a coherent memory structure. This integrative process enables the extraction of commonalities across experiences, thereby supporting generalization and inference-making[6,7]. In contrast, representational dissimilarities have been implicated in maintaining distinctions between related events, thus reducing memory interference[1–3]. The present study expands these previous findings by demonstrating the simultaneous encoding of integrated and separated representations for related events (see Fig. 5), each serving distinct memory demands. Specifically, we show that representational similarity enhances inferential memory performance by linking related events, whereas representational dissimilarity supports accurate source memory by preserving event-specific information. These findings refine our understanding of how the brain continuously modulates integration and separation in response to incoming information, offering insights into the neural mechanisms that balance flexibility and specificity in episodic memory formation.

The neural pattern similarities between AB and BC encoding were associated with both successful AC retrieval and impaired source memory (see Fig. 5). These results are in line with the integrative encoding account[4,35] and suggest that related events can be encoded into integrated representations that facilitate linking across event boundaries and support functions such as inferential reasoning and future decision making[6,7]. Importantly, our data suggests that this process comes at a cost: while memory integration facilitates inference, it is accompanied by reduced source memory. This aligns with prior evidence showing that integration often leads to a loss of episodic specificity, as overlapping experiences are blended into a more generalized representation[10,11]. Notably, the neural pattern dissimilarities between AB and BC encoding were associated with better source memory performance (see Fig. 5), indicating that memory separation efforts support the memory of specific event details[36,37]. Together, these highlight the idea that integration and separation mechanisms complement each other, allowing the brain to balance between forming generalized knowledge structures and maintaining distinct episodic memories. While integration aids inferential reasoning and decision by linking overlapping experiences, separation preserves the specificity of individual episodes and prevents interference.

Previous studies have reported that forming integrated memory representations impairs source memory and episodic details, implying that integrated and separated representations cannot coexist[10,11,26]. Our behavioral data show that AC retrieval is associated with better source memory, instead of a trade-off between memory integration and memory for specific events (Fig. 3B). Notably, however, the neural pattern similarities indicative of integration across AB and BC representations predicted reduced source memory accuracy (see Fig. 5). This finding aligns with the idea that integration across events may blur episodic details, making it more difficult to recall the contents of a given event[10,11,26]. Our results suggest that the brain forms both integrated and separated representations in parallel. While the formation of integrated representation may lead to a loss of detail, this does not prevent the formation of a parallel representation for the individual details (see also Supplementary Note 6). Taken together, the brain appears to simultaneously maintain integrated representations that facilitate indirect associations across related events, while retaining distinct representations for specific event details. Rather than competing with each other, our data suggested that the mechanisms underlying integrated and separated representations may operate independently.

The use of EEG allowed us to capture the temporal dynamics of these memory processes. Interestingly, neural similarities between AB and BC encoding were predominantly observed in the time windows containing the novel character C, consistent with the idea that the encounter with new information prompts its integration into an existing memory structure[6,7]. Nonetheless, similarity effects for character C presented in isolation versus C embedded within a context may reflect different underlying processes. Characters shown alone might evoke more abstract representations, reflecting a generalized agent that can appear across multiple interactions. By contrast, characters situated in vivid contexts are likely encoded in a context-specific manner, capturing the broader event structure and supporting memory for the episode[38,39]. Notably, only similarity effects observed during the context-rich segment predicted AC inference performance, raising the possibility that naturalistic and simplified stimuli shape memory representations in distinct ways. While this distinction may partly reflect the higher salience of animated, context-rich stimuli relative to static images, which could enhance the signal-to-noise ratio in similarity analyses, it also suggests that naturalistic paradigms provide a valuable window into how the brain represents episodic memories. In contrast to these neural similarities, neural dissimilarities between AB and BC encoding emerged when the previously seen character B was presented, suggesting that memory separation is triggered by the need to handle interference from overlapping information encountered across events[40,41]. Prior studies have shown that mnemonic interference is strongest when current experiences share high perceptual similarity with previously encoded memories[3,42]. Consistent with this view, the presentation of character B, common to both AB and BC movies, likely represents the moment of maximal interference, thereby eliciting neural dissimilarities. Tracking how these effects evolved over repeated BC encoding trials revealed incremental shifts in similarity and dissimilarity patterns, indicating a gradual development of the memory representations (see Supplementary Note 5). Repeated exposure to the movie clips likely strengthened memory traces of events, and in turn, facilitated the updating and refinement of both integrated and separated representations during the first few repetitions. Collectively, these incremental changes suggest that memory integration and separation are dynamic, experience-dependent processes that shape how memory representations evolve across related episodes.

Using univariate analysis, we further identified the TFR associated with encoding new events that share information with past experiences. By contrasting the TFR between BC and XY encoding, we found decreased alpha-beta power during the presentation of the novel character C and increased alpha-beta power during the presentation of the previously seen character B (see Fig. 6). Interestingly, these alpha-beta power modulations roughly corresponded to the time-windows where the neural pattern similarities and dissimilarities between AB and BC encoding was found in the RSA. Specifically, alpha-beta power decreases were observed during the time-window with pattern similarities, while alpha-power increases occurred in the time-window with pattern dissimilarities. A Bayesian regression analysis confirmed that alpha-beta power decrease predicted neural pattern similarities, whereas the alpha-beta increase predicted neural pattern dissimilarities. Previous studies have consistently reported the involvement of alpha-beta power decreases in memory encoding and retrieval[17–21,33,34,43], while alpha-beta power increases have been related with inhibitory control[23]. Our study suggests that alpha-beta power modulations are related to the formation of integrated and separated memory representations. The novel character C likely triggered the retrieval of the previously related AB event, leading to the integration of the character C into the existing memory representation. On the other hand, the increased alpha-beta power, observed when the shared character B was present, may have triggered inhibitory control, promoting the formation of separated memory representations.

In this study, structured movies with sequential presentation of event elements enabled the temporal dissociation of memory integration and separation. Our data showed that the engagement of these processes varies with the perceptual similarity of each segment to previously encountered overlapping events. Prior research has predominantly presented BC elements simultaneously, leaving the real-time formation of integrated and separated representations largely unexplored. Specifically, the overlapping element can reactivate prior memories and support integrated representations via hippocampal-neocortical interactions[4,35], whereas separation mechanisms may be recruited to resolve interference through prefrontal circuits that suppress competing representations[42,44]. Although integration and separation may operate in parallel, given their reliance on distinct neural substrates, the overlap in their temporal progression makes it difficult to distinguish between them[45]. This challenge may partly explain why previous studies of both rodents[46] and humans[8,42] typically reported either integration or separation. Examining specific hippocampal subregions is one strategy for identifying the coexistence of integrated and separated representations of overlapping events[9], but our structured movies provide an alternative approach to track the formation of these representations over time.

In conclusion, by revealing neural representational similarities and dissimilarities between related events, this study demonstrates that integrated and separated representations can both be formed during learning and coexist to support distinct memory functions. Learning across related events highlights the brain's capacity for adaptive memory processing, which is essential for generating novel inferences while preserving the fidelity of individual episodes. Future work should examine how distinct neural networks contribute to the formation of integrated and separated memories.

## Methods

### Participants

Following previous studies[9,47], we aimed for a sample size of 35 participants and finally recruited 41 participants. Data from one participant was excluded for not having at least one incorrect AC association trial. Additionally, the data from another four participants were excluded due to poor EEG data quality and/or experimental programming errors. Hence, the final sample consisted of 36 participants (27 female, 9 male, $M_{age} \pm SD = 24.3 \pm 2.94$).

Participants had normal or corrected-to-normal vision, were right-handed, fluent in English, and had no psychiatric or neurological disorders. Participants provided informed consent and were given a voucher that could be used in various shops for their time spent in the lab (on average 3 h per participant including EEG preparations). The data collection was anonymous and did not involve any potentially identifying demographic information. The data collection was conducted in accordance with the Swedish Act concerning the Ethical Review of Research involving Humans (2003:460) and the Code of Ethics of the World Medical Association (Declaration of Helsinki). As established by Swedish authorities and specified in the Swedish Act concerning the Ethical Review of Research involving Humans (2003:460), the present study does not require specific ethical review by the Swedish Ethical Review Authority due to the following reasons: (1) it does not deal with sensitive personal data, (2) it does not use methods that involve a physical intervention, (3) it does not use methods that pose a risk of mental or physical harm, (4) it does not study biological material taken from a living or dead human that can be traced back to that person. The Ethics Committee at the Department of Psychology, Lund University, has corroborated that the present research protocol follows the research ethics guidelines established by Swedish authorities.

### Stimuli material

AB and BC movies were generated using the life-simulation game The Sims 4 by Electronic Arts (www.thesims4.com). In each movie, two Sims were presented. The movies were 14 sec long and started with a fixation cross presented for 1 sec, followed by the presentation of the Sim A/C for 2 sec and the Sim A/C in a context, acting, for another 3 sec. After another 1-second fixation cross, the Sim B was presented for 2 sec followed by the Sim A/C interacting with Sim B in the same context for 5 sec (see Fig. 2A). Each movie has a unique context, so the AC association across AB and BC movies can be made via the overlapping element, i.e., Sim B.

In total, 48 AB movies and 48 BC movies were created. Each AB movie has its own corresponding BC movie. During the experiment, only 24 AB movies were presented during the AB encoding, while all the BC movies were displayed during the BC encoding. As a result, participants could recognize the Sim B in 24 BC movies, while in the other 24 movies, the Sims were new to the participants. These all-new movies are the XY movies, which were used to control for memory processes involved in encoding direct associations without any previous overlapping experience. To control for potential perceptual or narrative similarities between specific AB-BC pairs, the selection and presentation of AB movies were counterbalanced across participants. This ensured that all the movies presented in the second encoding round were equally likely to appear as BC or XY, minimizing confounds due to movie-specific similarities.

In addition to the experimental movies, three videos of a man jogging at different places, also generated in The Sims 4 environment, served as attention-check videos to keep participants concentrated. All participants responded to all attention-check videos.

To evaluate memory performance, 120 pictures of Sims faces (24 of each Sim A, B, C, X, and Y) were used as cues, targets, and distractors in the associative and source memory test. Additionally, to measure detail memory for context (see Supplementary Note 1), 96 pictures of contexts (48 for each AB and BC movie) were taken from the videos (without the Sim character present). One altered copy of each context was made by changing the color and/or texture details of the original context, which resulted in 96 pictures of contexts with altered details. To test detail memory for clothing, 96 full-body pictures of Sims (48 of each Sim A and Sim C) were used. The distractor pictures for the clothing test were generated by changing the color of the original clothing. For each Sim, three altered clothing pictures were generated, resulting in 288 pictures with altered details.

### Procedure

The stimuli were presented with PsychoPy[48] (v2022.1.0). The experiment consisted of an encoding and a test phase. In the encoding phase, participants first encoded 24 AB movies, each presented five times. When all 24 AB movies had been presented once, the next repetition began. After, participants encoded 24 BC movies intermingled with 24 XY movies, each presented also five times. The second repetition of these movies began when all of them had been presented one time. After the presentation of every 15 movies, the participants were encouraged to take a break. The attention-check movie was played three times during the encoding phase at random intervals. The participants were asked to press ENTER within three seconds each time they saw the attention-check movie. All included participants correctly responded to all attention checks within the given time.

The encoding phase ended with a short distraction task for 30 sec, where participants were asked to consecutively subtract 7 from a random 3-digit number. Then, the test phase started. The test phase began with the direct and indirect memory tests, which were intermixed, with the only constraint that the AC indirect memory test was made before the corresponding AB and BC memory tests. Both the direct and the indirect associative memory test started with a cue, corresponding to the face of the Sim A or C, displayed on screen for 1.5 sec, followed by the display of two faces presented below, the target Sim face C or A and the distractor Sim face drawn from the XY movies. Participants were asked to select which face was associated with the

cue by pressing the left or the right key. Immediately after that, memory for source was tested. Participants were asked to indicate if the two Sims had been seen together in the same movie. For both association and source memory tests, participants were asked to rate their confidence on a three-point scale: 1 - guessing, 2 - maybe, 3 - sure. After the association and source memory tests, a surprise episodic detail memory test was also implemented to detect the memory integration-related episodic detail loss (see Supplementary Note 1).

Using Bayesian $t$-tests, we confirmed that indirect AC association performance was equivalent whether A or C was the cue (Accuracy: difference = 0.039, $BF_{01}$ = 3.249; Response Time: difference = 0.027, $BF_{01}$ = 4.099 and Confidence: difference = 0.002, $BF_{01}$ = 4.113). This was performed with the BayesFactor package (0.9.12) in R (4.1.2), and $BF_{01}$ values were computed to quantify evidence for the null relative to the alternative hypothesis with gamma set to 0.707 by default. Following common conventions, $BF_{01}$ values greater than 3 were interpreted as providing moderate support for the null hypothesis[49,50].

## Behavioral data analysis and statistics
The behavioral data was analyzed in R (4.1.2) using linear mixed models. The packages of lme4 (1.1-34) and emmeans (1.8.7) were used to fit the models and perform statistical tests, and the effectsize (0.8.9) was used to estimate the effect sizes. The first level of each model was the trial level, which was clustered in the second level, i.e., the participant level. To rule out potential confounds of random effect[51] and simultaneously prevent model overfitting[52], the fitting of each model started with participants as the only random intercept. Other factors were thereafter added as random slopes in a step-by-step fashion. If an added random slope improved the model fitting significantly (i.e., $p < .05$ in chi-square test for model fitting comparison, decreased Akaike Information Criterion and Bayesian Information Criterion), the random slope was kept. The final equations for all models, showing which random factors were considered, can be consulted in the Supplementary Note 2. For each model, the homogeneity of variance of the residuals was assessed by using Levene's test. If the test indicated heteroskedasticity, we re-ran the model with a restricted variance structure[28,53]. For inferential statistics, we used two-tailed tests with the significance criterion of $p < .05$. Correct responses outside a 10-sec time window or with a confidence rating of 'guessing' were considered incorrect to prevent responses based on logical reasoning rather than memory from biasing the results. Only correct responses were considered in the response times and confidence analysis. Response times were logarithmic transformed to correct for skewness.

First, behavioral performance for indirect and direct associative memory was contrasted in terms of accuracy, response times (RTs), and confidence ratings, with Association Type (AC vs AB vs BC vs XY) as the predictor for all three models. Next, source memory accuracy and confidence were also examined using Association Type (AB vs BC vs AC vs XY) and Association Accuracy (correct vs incorrect) of the preceding memory test as predictors. To interpret the interaction term observed in the source memory models, Monte Carlo sampling was implemented with in-house MATLAB scripts to estimate the effect size differences across all Association Types. For each Association Type, the effect sizes of the Association Accuracy over source memory accuracy and confidence were sampled for 10000 iterations, based on the model estimates obtained above. Then, these effect sizes were compared across Association Types by estimating the distribution of the difference between these samples. The percentage (%) of the sample difference that is larger/smaller than zero was considered an approximate of the one-tail statistical probability, which was then multiplied by two to indicate two-tail $p$-value, where the 0.05 criterion was applied to determine significance.

The normality of residuals was assessed by inspecting the Q-Q plots of the standardized residuals and Shapiro-Wilk's tests. The homogeneity of variance of the model residuals was assessed for each

model by visually inspecting a plot of the model residuals versus fitted values and using Levene's test for unequal variance. Significant complex effects were followed up with post hoc tests with Tukey correction. Effect sizes, partial eta squared ($\eta_p^2$) for F-tests and unstandardized difference ($D$) for t-tests, and 95% confidence intervals (CI) were also reported together with other statistics.

## EEG data collection and preprocessing
EEG was recorded using a SynAmps RT Neuroscan 64-channel amplifier (sampling rate 1 kHz, bandwidth DC-3500Hz, 24-bit resolution, left mastoid reference) with 62 electrodes attached to an elastic cap (active electrode EasyCap). The cap was placed according to the extended 10–20 system. Furthermore, an electrode was attached to the skin under the participant's left eye to detect and later filter noise caused by eye blinks.

To preprocess the data, the EEG waveform was downsampled to 500 Hz, and epochs of 14 sec, ranging from -0.5 to 13.5 sec with respect to the onset of each movie, were created. Preprocessing was implemented with FieldTrip[32] accompanied by in-house scripts. The data were low-pass filtered at 200 Hz. A notch filter at 50, 100 and 150 Hz was applied to remove Alternate Current related signal. The data was transformed to linked-mastoid reference and demeaned. Vertical eye movements and blinks were estimated using the electrodes placed under the left eye and Fp1 in the cap, and horizontal eye movements were estimated using the Fp9 and Fp10 electrodes. The data were then manually checked to detect noisy channels and to remove trials dominated by artefacts other than eye blinks and horizontal eye movements. An independent component analysis was then applied to detect and remove components related to blinks and horizontal eye movements. Finally, after interpolating the removed channels, another visual check-up of the data took place for a final rejection of trials with residual artifacts. The cleaned data resulted, on average per participant, in 114 AB trials (ranging from 102 to 120), 114 BC trials (ranging from 100 to 119) and 114 XY trials (ranging from 107 to 119).

## Time-Frequency decomposition of EEG data
The time-frequency decomposition was also implemented with in-house MATLAB scripts based on the Fieldtrip package[32]. The Morlet wavelets[54,55] with 5 cycles were used to extract power spectra of frequencies of 3–30 Hz (28 frequencies in total) for each event encoding trial (AB/BC/XY) with a temporal resolution of 0.02 sec.

## Feature selection for the time-resolved representational similarity analysis
Before performing the RSA, we aimed to first identify the time-frequency features that were sensitive to AB movie content. To account for potential temporal variations across repeated viewings of the movies[30], we employed a wavelet coherence approach. Wavelet coherence measures the correlation between the spectral components of two time-series, which is highly sensitive to subtle phase shifts[29]. It can be computed at the full channels × timepoints × frequency bands level for both within- and between-movie comparisons. This preserves the multidimensional structure of the EEG data, making it well-suited for a fine-grained feature stability assessment. By evaluating the coherence between all repetitions of the same AB movie and contrasting it against the coherence between different AB movies, we identified the time-frequency features sensitive to AB movie content. These features were later used to assess the representational similarities in the neural data between AB and BC/XY movies.

Wavelet coherence was estimated at the level of individual channels × timepoints × frequency bands. For each participant, we calculated the pairwise wavelet coherence across all five repetitions of each AB movie, resulting in ten unique within-movie repetition pairs per AB movie. As a control, we computed the wavelet coherence between each AB movie and all other AB movies across all five repetitions,

resulting in 230 pairs per movie. For each participant, the within-AB movie coherence was obtained by averaging the wavelet coherence across 240 different pairs (24 different AB movies × 10 repetition pairs) and the between-AB movie coherence was obtained by averaging the coherence across 5520 pairs (24 different AB movies × 23 other AB movies × 10 repetition pairs). This yielded two separate 3-dimensional coherence matrices (channel × timepoint × frequency) per participant: one for within-AB movie coherence and one for between-AB movie coherence.

A cluster-based permutation test was used, at group level, to contrast the within-AB movie coherence against the between-AB movie coherence across all participants[56]. The test was conducted separately for four distinct time windows, corresponding to the different segments of the AB movie, i.e., 'Sim A', 'Sim A in Context', 'Sim B', and 'Sim A and B in Context'. For all comparisons, $t$ values of significant clusters (two-tail $p$ value ≤ .05), as well as their corresponding $p$ values, are reported. Significant clusters indicated the time-frequency features sensitive to AB movie content, which were then used as templates to extract features (i.e., power at each channel-timepoint-frequency) for the representational similarity analysis.

## Time-resolved representational similarity analysis and statistics

The representational similarity between the neural patterns associated with AB encoding and the BC/XY encoding was estimated as a measure of the approximation and differentiation of mental representations[57]. Accordingly, similarities in the neural representation are indicators of memory integration linking the events, while dissimilarities reveal memory separation between the events[9].

The similarities between AB and BC/XY movies were estimated for each participant and movie. TFR was transformed into decibel (dB) values relative to the average power across the epoch. This approach reduces the disproportionate influence of low-frequency power and emphasizes relative power, rather than absolute power, fluctuations in relation to specific movie contents. For each AB movie, a 3-dimensional representative channel-timepoint-frequency representation was obtained by averaging across the five repetitions. Representative features for each AB, BC and XY movie segment were extracted using the template obtained in the wavelet coherence analysis. Considering the time shifts across repetitions[30], the medians along the time dimension of the extracted features were used as a stable estimation. This procedure resulted in a 2-dimensional (channel-frequency) feature map for each AB movie segment, which was then reshaped into a feature vector by concatenating the values of each channel head to tail, representing the power distribution in the 'channel-frequency' space of a segment. In total, this feature extraction procedure resulted, for each participant, in 96 feature vectors of AB movies (24 AB movies × 4 different AB movie segments) to be used in the similarity analysis contrasting the neural representations of AB and BC/XY movies. Feature vectors of BC and XY movies were extracted with the same template at each timepoint of the whole epoch to obtain the time-resolved similarities.

Pearson correlation coefficients were used to quantify the neural similarity and were Fisher z-transformed for statistical analysis. The correlations between AB movies and BC/XY movies were performed separately for each of the five BC/XY repetitions, which enabled us to account for the progression of the similarities and dissimilarities along repetitions. The feature vectors corresponding to the AB movie contents were correlated with each timepoint of the BC/XY movie, resulting in a time-resolved representational similarity time series. For each BC/XY timepoint, the 2-dimensional matrix (channel-frequency) of power was reshaped in the same way as the extracted AB feature map, resulting in a vector representing the power distribution in the 'channel-frequency' space of each timepoint. This vector was correlated with the feature vector of each AB movie segment, resulting in 96 different correlation values (24 AB movies × 4 different segments),

indicating the representational similarity between each segment of AB movie and each timepoint of each BC/XY movie. The time-resolved topographical distribution of the similarity effects was assessed by calculating the Pearson correlation of each channel and its neighbors at each timepoint (see references[27,58] for a similar approach). The results of identified time windows are summarized in Supplementary Note 4. In the main paper, we focus on the similarity between the segment when Sim A and B interact in a context, as this segment contains all the relevant information and encompasses the AB movie event. The representational similarity analysis involving the other AB movie segments is presented in Supplementary Note 7.

The statistical evaluation of the RSA was performed with Bayesian statistics for its improved sensitivity, greater robustness to noise, and more effective control of multiple comparisons[59–61]. To account for false discoveries, we applied an informative Normal-Inverse-Gamma distribution prior (alpha = 15, beta = 15, mu = 0, v = 30)[27,59,62]. The posterior was obtained by updating the prior with the data of all participants consecutively and tested against 0. Following conventional interpretation, a Bayesian Factor ($BF_{10}$) exceeding 3 was taken as statistical criterion, indicating moderate support for the alternative hypothesis against null hypothesis[49,50]. Only similarities and dissimilarities last over 0.1 s were reported to avoid spurious results.

Note that when participants were first exposed to Sim C in the BC movies at repetition zero, we did not expect any memory reactivation of the AB movie. Thus, we excluded the repetition zero of the BC/XY trials from both the representational similarity analysis and the univariate analysis.

## Relationship between behavioral performance and representational similarities

To further reveal the behavioral consequences of representational similarities and dissimilarities between AB and BC, we investigated the relationship between the (dis-)similarities and memory performance.

Bayesian linear regression with a Multivariate-Normal-Inverse-Gamma distribution prior (alpha = 15, beta = 15, mu = 0, lambda = 30 * $I$, $I$ is unit matrix) and the data at trial level was used in the present analysis. We estimated standardized regression coefficients to control for individual differences, using a $BF_{10} > 3$ as the statistical criterion[49,50]. For each trial, the (dis-)similarities within each time window of significance were extracted and averaged as dependent variables. To estimate the (dis-)similarity difference between trials with correct and incorrect AC association, the accuracy of AC association was used as the independent variable, with the repetition order included as the controlled variable. Similarly, to estimate the (dis-)similarity difference between trials with correct and incorrect source memory for AC association, the accuracy of source memory was used as the independent variable, with the repetition order and the AC association accuracy included as the controlled variables. To ensure that the results were not biased by unbalanced data, the present analyses included only the participants with at least 10% incorrect trials for the independent variable (i.e., accuracy for AC association or source memory ≤ 0.9). Hence, the analysis for AC association included 30 participants, while the analysis for the source memory included 28 participants. A similar analysis for the clothing memory was also performed; see Supplementary Note 8 for details.

## EEG univariate analysis and statistics

A univariate analysis was used to investigate the neural correlates associated with the encoding of new events that share information with past events. To assess absolute power differences between BC and XY encoding, we averaged the spectra across all BC and XY trials, respectively, and computed their difference through direct subtraction. This difference was then scaled by the power of BC for each participant to control for individual variability and ensure comparability across participants, i.e., (BC-XY)/BC, and the statistical tests

were performed by contrasting this index against zero using the cluster-based permutation test[56]. The comparison was performed separately for each segment of the movie (i.e., 'Sim C' vs. 'Sim Y', 'Sim C in Context' vs. 'Sim Y in Context', 'Sim B' vs. 'Sim X', and 'Sim C and B in Context' vs. 'Sim Y and X in Context') using a cluster-based permutation analysis implemented in fieldtrip[32]. For all comparisons, $t$ values of significant clusters (two-tailed $p$ value ≤ .05), as well as their corresponding $p$ values, are reported.

Next, we evaluate if the potential differences between BC and XY encoding were predictive of the similarities and dissimilarities. This was performed using a Bayesian linear regression approach with a Multivariate-Normal-Inverse-Gamma distribution prior (alpha = 15, beta = 15, mu = 0, lambda = 30 * $I$, $I$ is unit matrix) and the data at trial level. The power / similarity values were extracted from the trials according to the cluster / time-window of significance. Standardized regression coefficients were estimated to reduce the influence of individual differences, and the coefficient with $BF_{10} > 3$ was considered as moderate evidence for alternative hypothesis against null hypothesis[49,50].

### Reporting summary
Further information on research design is available in the Nature Portfolio Reporting Summary linked to this article.

## Data availability
The behavioral and EEG data presented in this study are available in the Zenodo database (https://doi.org/10.5281/zenodo.17612987)[63]. To support reproducibility, we have shared preprocessed datasets, providing a straightforward starting point for analysis. The EEG raw data underlying this study is part of an ongoing project with additional analyses planned through 2028. The full, anonymized dataset will be deposited in Zenodo database and made publicly accessible in January 2028.

## Code availability
The code used in the present study is available in the Zenodo database (https://doi.org/10.5281/zenodo.17612987)[63].

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

## Acknowledgements

This work was supported by the Swedish Research Council Grant VR 2019-02455 awarded to Inês Bramão. We thank Kira Friedrichs for the assistance in data collection and stimuli preparation and all the volunteers who participated in this study. The computations were enabled by resources provided by LUNARC, The Center for Scientific and Technical Computing at Lund University.

## Author contributions

Z.L., M.J. & I.B.–Conceptualization and Methodology; Z.L.–Investigation, Data Curation and Formal Analysis; Z.L. & I.B.–Visualization, Writing: Original Draft Preparation; Z.L., M.J. & I.B.–Writing: Review & Editing; I.B. & M.J.– Supervision and Funding Acquisition.

## Funding

## Competing interests

The authors declare that there are no competing interests.
