## [Transparent Peer Review file · Nature Communications]

Episodic events are flexibly encoded in both integrated and separated neural representations

Corresponding Author: Dr Zhenghao Liu

Version 0:

Reviewer comments:

Reviewer #1

(Remarks to the Author)

This study examines how the brain flexibly encodes episodic events to support different memory functions. Using EEG, the authors recorded neural activity while participants watched short movie clips depicting overlapping social interactions. Memory was later tested for direct, indirect, and source associations. Behaviorally, successful inference of indirect associations was linked to more accurate source memory. Time-resolved representational similarity analysis revealed that neural pattern similarity during encoding supported memory integration and indirect inference, while dissimilarity supported accurate source memory by preserving distinct event representations. These findings suggest that the brain encodes episodes in both integrated and segregated formats to meet different memory demands.

I found the study to be both interesting and well-executed. It combines several state-of-the-art methodological approaches in a compelling way and offers a valuable opportunity to examine complex memory representational dynamics that are rarely explored in such a temporally rich and ecologically valid context. Overall, I believe the manuscript has the potential to make a meaningful contribution to the field. I offer the following suggestions to enhance methodological clarity and interpretability.

Majors:

1. More detail is needed regarding the computation of coherence. It is important to clarify whether coherence was computed at the individual electrode \times time \times frequency level and then averaged across channels, or whether it was aggregated in a different way. Additionally, it would be helpful to know at what point the permutation analysis was implemented relative to these steps.
2. The normalization approach used in the time–frequency representations (TFRs) warrants clarification. The RSA was based on power changes relative to the average power across the entire epoch, whereas the univariate power analysis in Figure 6 appears to use a baseline correction method tied to the time window immediately preceding each target segment. This likely explains the presence of NaN values during the baseline period. If different normalization methods were applied to the RSA and univariate power analyses, this should be explicitly justified. Otherwise, it may be advisable to re-evaluate the univariate power analysis using the same normalization strategy applied in the RSA to ensure consistency.
3. Were the same features extracted from AB movies also used in the BC RSA? I assume so in order to make vector-based correlation feasible, but it should be explicitly stated to avoid confusion.
4. Relatedly, the type of correlation metric used in the RSA should be reported. Was Pearson or Spearman correlation used? If Pearson, were r -values transformed into z -scores before group-level statistical analyses to ensure comparability across distributions?
5. On a more conceptual note, the rationale for using coherence as the central neural feature should be addressed more directly. Why was coherence chosen over more standard power-based TFR features? Would the same representational patterns emerge with power instead of coherence? As this appears to be one of the first studies to use coherence in this way, it would be helpful to highlight its novelty and justify its use over other possible measures.

Minors:

1. It would be useful to briefly describe the random effects structure of the mixed-effects models (e.g., whether by-subject or by-item intercepts or slopes were included).
2. There is a typo in the section describing the behavioral data analysis: "Tucky-correction" should be corrected to "Tukey correction."

Reviewer #2

(Remarks to the Author)

The manuscript by Liu and colleagues describes an EEG study in which 36 participants viewed movies simulating real-life interactions. In the experiment, two sets of movies (AB and BC pairs) featuring a shared character were used to investigate the temporal dynamics of forming integrated and segregated memory representations and their relationship with subsequent memory accuracy. Behaviorally, they found that source memory performance was positively correlated with AC memory inference performance. Using time-resolved representational similarity analysis (RSA), they found that the neural representations of AB and BC movies share both similarities and dissimilarities. During the encoding of the novel character C, there were neural similarities to the AB representation, which predicted successful AC memory inference. In contrast, during the presentation of the common character B, neural dissimilarities to the AB representation were observed, and these predicted accurate source memory. Lastly, univariate analyses revealed that changes in alpha-beta power were associated with these neural (dis)similarity patterns. In summary, the authors provide compelling evidence that the brain dynamically and flexibly engages in both memory integration and separation during the encoding of overlapping events. These processes have distinct temporal and electrophysiological signatures and support different, complementary memory functions.

Overall, I find this study is well-motivated and clearly written. The analyses are appropriately conducted and convincing. The findings expand our understanding of the neural mechanisms for forming and maintaining both integrated and segregated memory representations. In particular, temporally separating the neural processing of integration and separation is novel (to my knowledge) and has broad implications for our understanding of the neural mechanisms underlying overlapping memories. I only have a few comments for the authors, which are mostly about the interpretation of the results:

1. The neural similarity analysis revealed increased neural similarity between the Sim A&B representation and the Sim C (and Sim C in context) representation. The authors interpret this as evidence of memory integration. I wonder whether it could merely reflect the reactivation of the Sim A&B memory, rather than the formation of an integrated A-B-C representation.
2. The main analyses focused on the neural similarity between the representation of Sim A&B in context and the BC trials. I am somewhat surprised that there was little to no evidence showing similarity to Sim A or Sim B individually during the Sim C segment. More importantly, there appears to be no Sim B-specific similarity during the Sim B and Sim B&C in context segments, where character B was present on screen. This makes the interpretation of neural similarity ambiguous, as the analysis appears unable to capture common visual inputs.
3. Related to the previous comment, I wonder what the neural dissimilarity truly reflects. The authors argue that this dissimilarity is evidence of the segregation of AB and BC memories. If that is the case, why would this dissimilarity appear in the Sim B segment instead of the Sim B&C in context segment, where the full BC memory event is arguably most salient?
4. The results show two significant similarity clusters corresponding to the Sim C and Sim C in context segments. Do they reflect the same underlying neural process? More generally, what is the functional difference between the character segment and the character-in-context segment? After the initial learning, can the character segment (e.g., Sim C) alone trigger memory integration, or is the context an essential component for forming overlapping memories?
5. The current experimental design nicely separates the memory integration and separation processes in time and examines their relationship to behavior. To bridge the results to the existing literature, it might be worthwhile to discuss the possible mechanism if the BC pair were presented simultaneously (i.e., only a Sim B and C in context segment). Would integration and separation still both occur? If so, would they have different temporal dynamics within that single segment? Or might the two processes occur simultaneously but be supported by different neural circuits?
6. Given that both integration and separation processes are observed, I wonder if there is any evidence of a trade-off between them. For example, could stronger integration of a BC memory into an AB memory lead to a weaker separation process in the Sim B segment, which might, in turn, affect source memory performance? Conversely, could stronger integration create a greater need for separation, thereby increasing the observed neural dissimilarity? While this is an interesting question, I would understand if the authors find it to be outside the scope of the current paper.
7. The RSA results showed that the neural similarity and dissimilarity measurements predict AC inference and source memory performance. Are these neural markers also associated with memory for episodic details? For example, might a strongly integrated representation come at the cost of memory for details like the character's clothing? Again, I understand if the authors consider this question beyond the scope of the current paper.

Reviewer #3

Summary

This study makes a timely and valuable contribution to the field of memory research by investigating how the brain encodes episodic events to support multiple mnemonic functions. Leveraging an ABC associative inference paradigm—a framework well-suited to studying memory generalization and inference—the authors employed naturalistic stimuli (simulated real-life movies) and multivariate analysis techniques to explore the flexible encoding of related experiences.

Participants first watched a series of short movies depicting interactions between two characters (AB pairs), followed by a second set of scenes involving one familiar and one novel character (BC pairs). This design enabled the testing of indirect (AC) associations, as well as source memory—i.e., whether participants remembered if two characters had ever appeared together. This dual focus allowed the authors to probe two distinct but complementary memory processes: integration, where overlapping events are linked into unified representations, and separation, where distinct experiences are preserved independently.

The use of time-resolved representational similarity analysis (RSA) on EEG data allowed the authors to track the neural dynamics underlying memory formation with high temporal precision. Importantly, the results revealed that neural pattern similarity between AB and BC episodes predicted successful inference of indirect (AC) associations—indicating integration. In contrast, neural dissimilarity predicted accurate source memory, pointing to the preservation of separate memory traces. This double dissociation is both conceptually and methodologically interesting. It suggests that the brain does not rigidly encode experiences in one way but instead adapts flexibly, balancing integration and separation depending on the demands of future retrieval. These findings align with and extend theoretical frameworks suggesting that memory systems must support both generalization (e.g., for inferential reasoning) and specificity (e.g., for accurate source recollection). The use of naturalistic stimuli adds ecological validity, enhancing the relevance of the findings for real-world memory processes. Moreover, the combination of behavioral and neural data—analyzed with state-of-the-art multivariate methods—demonstrates a sophisticated experimental approach that provides insights not accessible with traditional univariate or trial-averaged analyses.

While the presented findings add to a growing body of evidence with potential implications for memory disorders, I have some major and a few minor issues that would need to be addressed and clarified before publishing. I detail these concerns below.

Major concerns:

1. My primary concern lies with the introduction, which needs to be substantially rewritten to better frame the study's rationale. As it stands, the introduction heavily emphasizes prior fMRI findings, particularly the role of hippocampal subregions in ABC inference paradigms. While this context is somewhat relevant, it reads more like a retrospective interpretation of the current results than a focused theoretical or empirical motivation for the hypotheses tested in this EEG study. Critically, the introduction fails to reference existing M/EEG research that has examined inference and generalization using similar paradigms, including studies employing representational similarity analysis (RSA). These omissions weaken the justification for using EEG and limit the reader's understanding of how this work builds on or extends prior findings especially concerning brain oscillations.
2. Furthermore, although the authors perform both univariate and multivariate time-frequency analyses, the introduction lacks any predictions of relevant frequency bands—such as the θ —which are known to play a key role in memory integration and retrieval. Without grounding the study in this literature, the introduction does not adequately set up the relevance of frequency-specific dynamics to the research questions. A revised introduction should clearly articulate the motivation for using EEG, specify what temporal or spectral insights it can offer beyond fMRI, and situate the work within the existing electrophysiological literature on memory generalization and inference.

Generally, the results seem incomplete and demand deeper analyses:

3. A major concern relates to the spatial characterization of the reported neural similarity effects. While I appreciate that the RSA was conducted at the whole-brain level, the results as presented lack any indication of where on the scalp these effects are most prominent. Especially given the introduction's emphasis on specific brain regions and memory-related processing, I would have expected at least a basic breakdown of whether the observed effects are stronger over frontal, parietal, or other regions. To enhance interpretability, I recommend the authors either (1) report the topographical distribution of the similarity effects or (2) re-run the RSA restricted to functionally meaningful electrode clusters (e.g., frontal, central, parietal). This would provide valuable insight into the potential cognitive and neural sources of the effects and better connect the findings to prior literature on memory integration and segregation.

4. Closely related to the previous point, it is unclear why no source reconstruction analyses were conducted, especially given that the experiment was designed around RSA with a clear theoretical focus on specific brain regions, notably the hippocampus. While I understand that the RSA was performed at the sensor level, this approach limits anatomical interpretability—particularly in a study aiming to link EEG pattern similarity to memory mechanisms commonly associated with deep brain structures. Given the relevance of medial temporal regions in memory integration and inference, I strongly recommend that the authors complement their current analyses with source-level RSA or at least conduct source localization

of the key effects. This would significantly strengthen the anatomical specificity of the findings and align the EEG results more directly with the fMRI literature that motivated the study.

5. Given the well-known limitations of detecting hippocampal activity with EEG—even when employing advanced source reconstruction methods such as LCMV beamformers—I believe it would be valuable for the authors to explore functional connectivity of relevant cortical regions, such as the angular gyrus, in relation to the reported effects. Incorporating connectivity analyses would enhance the interpretive depth of the findings and provide a more comprehensive picture of the neural mechanisms underlying the observed behavioral outcomes.

Minor concerns:

1. It would be helpful if the authors addressed potential effects of the repeated presentation (5×) of the stimuli. Since neural signals were averaged across repetitions, the role of novelty vs. familiarity—a key factor in memory encoding—should be considered. The neural dynamics of the first versus the fifth presentation may differ substantially, and this could impact both the behavioral and neural findings. A brief analysis or at least a discussion of how repetition may have influenced RSA results are needed.

2. The behavioral data in Figure 3 appear to exhibit substantial ceiling effects, particularly in accuracy scores. The authors should clarify whether the use of linear mixed models (LMMs) is appropriate given the limited variance. Please elaborate on how this was handled in the analysis and interpretation.

3. Response times for BC retrieval were notably slower than for AB. This difference deserves further discussion. Could this be due to fatigue, re-encoding demands, or greater cognitive load during BC trials? Including a plausible interpretation would help contextualize this finding.

4. In the caption of Figure 2, there is a minor typo:

“(D) Systematic similarities, indicative of memory integration should predict of AC retrieval performance.”

This should be corrected to:

“(D) Systematic similarities, indicative of memory integration, should be predictive of AC retrieval performance.”

5. Did the authors assess the perceptual or narrative similarity between the SIM movies and stories (AB, BC pairs)? If some stimuli were more similar than others, this could confound RSA results. It would strengthen the study to report whether similarity ratings (e.g., by independent raters) were obtained and whether these were controlled for or included in the analyses.

Version 1:

Reviewer comments:

Reviewer #1

(Remarks to the Author)

The authors did an excellent job and have fully addressed all my previous concerns. I am happy to recommend the manuscript for publication in its current form.

Reviewer #2

(Remarks to the Author)

The authors have satisfactorily addressed all my comments. I found the supplementary analyses and additional discussions very useful. I have no further concerns and am happy to recommend the manuscript for publication.

Reviewer #3

(Remarks to the Author)

The authors have substantially revised the manuscript and have provided thorough, detailed responses to all of my questions. The Introduction has been carefully revised as requested and is now significantly improved in both clarity and structure. All of the concerns I previously raised have been fully addressed, and I now consider the manuscript to be in excellent shape.

RESPONSE LETTER TO REVIEWER COMMENTS

Reviewer #1 (Remarks to the Author):

This study examines how the brain flexibly encodes episodic events to support different memory functions. Using EEG, the authors recorded neural activity while participants watched short movie clips depicting overlapping social interactions. Memory was later tested for direct, indirect, and source associations. Behaviorally, successful inference of indirect associations was linked to more accurate source memory. Time-resolved representational similarity analysis revealed that neural pattern similarity during encoding supported memory integration and indirect inference, while dissimilarity supported accurate source memory by preserving distinct event representations. These findings suggest that the brain encodes episodes in both integrated and segregated formats to meet different memory demands.

I found the study to be both interesting and well-executed. It combines several state-of-the-art methodological approaches in a compelling way and offers a valuable opportunity to examine complex memory representational dynamics that are rarely explored in such a temporally rich and ecologically valid context. Overall, I believe the manuscript has the potential to make a meaningful contribution to the field. I offer the following suggestions to enhance methodological clarity and interpretability.

We thank the reviewer for the positive and constructive remarks about our study and paper. We carefully considered the thoughtful suggestions to enhance methodological clarity and interpretability, and the manuscript has been revised accordingly. We are grateful for these valuable insights, which have significantly strengthened the quality of the work.

Majors:

1. More detail is needed regarding the computation of coherence. It is important to clarify whether coherence was computed at the individual electrode \times time \times frequency level and then averaged across channels, or whether it was aggregated in a different way. Additionally, it would be helpful to know at what point the permutation analysis was implemented relative to these steps.

We thank the reviewer for noticing that information was missing regarding the computation of coherence. The pair-wise coherence of AB movies was estimated at each channel \times timepoint \times frequency level. Specifically, for each participant, the within-AB movie coherence was obtained by averaging the coherence across 240 different pairs (24 AB movies \times 10 repetition pairs), while the between-AB movie coherence was obtained by averaging the coherence across 5520 pairs (24 AB movies \times 23 different AB movies \times 10 repetition pairs). This provided the coherence estimation for each participant. To assess the overall difference in coherence between within-AB and between-AB movie conditions across participants, we conducted a permutation test to identify clusters of channel \times timepoint \times frequency showing higher coherence for the within-AB movie condition compared to the between-AB movie condition. These clarifications are now added to the text (see page 29, paragraph 1 & 2).

2. The normalization approach used in the time–frequency representations (TFRs) warrants clarification. The RSA was based on power changes relative to the average power across the entire epoch, whereas the univariate power analysis in Figure 6 appears to use a baseline

correction method tied to the time window immediately preceding each target segment. This likely explains the presence of NaN values during the baseline period. If different normalization methods were applied to the RSA and univariate power analyses, this should be explicitly justified. Otherwise, it may be advisable to re-evaluate the univariate power analysis using the same normalization strategy applied in the RSA to ensure consistency.

We thank the reviewer for noticing the lack of clarity regarding the normalization approaches applied to the data. Importantly, both the RSA and univariate analyses used the same initial baseline correction: a mean subtraction (demeaning) across the entire epoch prior to the time-frequency decomposition. No additional pre-stimulus baseline correction was applied. The NaN values observed in the pre-stimulus interval of the univariate time-frequency plots result from edge effects inherent to the wavelet decomposition and not from baseline correction.

However, due to the distinct analytical goals of the two analyses, we applied different scaling approaches, each optimized for its respective purpose: detecting relative power differences across conditions in the RSA, and absolute power differences across conditions in the univariate analysis.

For the RSA, to emphasize relative power differences rather than absolute, we applied a decibel (dB) transformation relative to the average power across the entire epoch. This approach is widely used in EEG-RSA (e.g., Pacheco Estefan et al., 2019) and mitigates confounds introduced by the 1/f distribution, which would otherwise cause low-frequency power to dominate similarity estimates. Both pre-stimulus and whole epoch references have been suggested for such scaling (Grandchamp & Delorme, 2011; Hu et al., 2014), each with advantages and disadvantages. A pre-stimulus window avoids contamination from stimulus-evoked activity, but it may suffer from variability across trials, potentially reducing reliability (Hu et al., 2014). In contrast, a whole-epoch reference provides greater stability but may incorporate evoked responses, potentially biasing absolute power estimates (see also Grandchamp & Delorme, 2011). Given our experimental design and trial structure, we selected the whole-epoch average as the reference. Although this choice may include evoked activity, RSA is based on Pearson correlations, which demean the feature vectors and therefore emphasize relative rather than absolute power. Consequently, any bias in absolute power is unlikely to substantially impact the RSA results.

For the univariate analysis (Figure 6), the goal was to examine absolute power differences across conditions. Applying the same dB transformation as in the RSA would have distorted these estimates. Instead, we compared conditions by subtracting XY from BC and scaling by BC power to account for individual differences.

Importantly, the correlation analysis (see page 17, paragraph 2) shows that the power differences observed in the univariate analysis correspond to the similarities and dissimilarities observed in the RSA. This further supports our methodological choice and indicates that the observed time-frequency effects reflect processes related to memory integration and separation rather than trivial scaling artifacts.

These clarifications have been added to the text (see page 29, *METHOD* → *Time-Resolved Representational Similarity Analysis and Statistics*, paragraph 2; and page 32, *METHOD* → *EEG Univariate Analysis and Statistics*, paragraph 1).

3. Were the same features extracted from AB movies also used in the BC RSA? I assume so in order to make vector-based correlation feasible, but it should be explicitly stated to avoid confusion.

We thank the reviewer for this helpful comment. The reviewer is correct that the features extracted from the AB movies were also used in the BC/XY RSA, ensuring that the vector-based correlation was feasible. This has now been explicitly clarified in the manuscript (see page 30, top paragraph continuing from the previous page).

4. Relatedly, the type of correlation metric used in the RSA should be reported. Was Pearson or Spearman correlation used? If Pearson, were r -values transformed into z -scores before group-level statistical analyses to ensure comparability across distributions?

We thank the reviewer for noticing that this important methodological detail was absent in the original text. The correlation metric used in the RSA was the Pearson correlation coefficient, and the resulting r -values were Fisher z -transformed prior to subsequent averaging and subtraction. This information has now been added to the manuscript (see page 30, paragraph 1).

5. On a more conceptual note, the rationale for using coherence as the central neural feature should be addressed more directly. Why was coherence chosen over more standard power-based TFR features? Would the same representational patterns emerge with power instead of coherence? As this appears to be one of the first studies to use coherence in this way, it would be helpful to highlight its novelty and justify its use over other possible measures.

We thank the reviewer for raising this point, as it provides an opportunity to clarify our feature selection strategy. Since each movie was presented five times, temporal shifts in neural responses are expected to occur across the repeated viewings (Lee et al., 2021). Wavelet coherence is particularly sensitive to subtle phase differences (Lachaux et al., 2002), making it a well-suited tool to capture and quantify these temporal misalignments.

An additional advantage of coherence is that it can be computed at the full channel \times timepoint \times frequency level for both within- and between-movie comparisons, thereby preserving the multidimensional structure of the data. In contrast, power-based TFR features cannot be used in this way without losing at least one dimension, which would limit their capacity to capture both within- and between-AB movie consistency.

Accordingly, we used wavelet coherence to extract features that reflected stable neural representations for each AB movie by selecting those features that were consistent across repetitions (within-AB movie) yet distinct across different movies (between-AB movie). This rationale has now been explicitly described in the manuscript (see page 11, *RESULTS* \rightarrow *EEG Results* \rightarrow *Feature Selection Reveals Time-Frequency Signatures of AB Movie Content*, paragraph 1; and page 28, *METHOD* \rightarrow *Feature Selection for the Time-Resolved Representational Similarity Analysis*, paragraph 1).

Minors:

1. It would be useful to briefly describe the random effects structure of the mixed-effects models (e.g., whether by-subject or by-item intercepts or slopes were included).

We thank the reviewer for this suggestion. The random effect structure for each model is described in detail in the supplementary materials (see **Supplementary Note 2**). To rule out potential confounds of random effect (Ambrosius, 2007) while avoiding model overfitting (Matuschek et al., 2017), we adopted a stepwise approach. Model fitting began with

participants as the only random intercept, after which additional factors were sequentially added as random slopes. A random slope was retained only if it significantly improved model fit (i.e., $p < .05$ in chi-square test for model fitting comparison, decreased Akaike Information Criterion and Bayesian Information Criterion). For each model, we also assessed the homogeneity of variance of the residuals using Levene's test. In cases of heteroskedasticity, we re-ran the model with a restricted variance structure (Audrain & McAndrews, 2022; Liu et al., 2024).

These details have now been added both to the supplementary materials (see **Supplementary Note 2**) and to the manuscript (see page 26, *METHOD* → *Behavioural Data Analysis and Statistics*, paragraph 1).

2. There is a typo in the section describing the behavioral data analysis: “Tucky-correction” should be corrected to “Tukey correction.”

We thank the reviewer for noticing this typo. The text has been corrected.

Reviewer #2 (Remarks to the Author):

The manuscript by Liu and colleagues describes an EEG study in which 36 participants viewed movies simulating real-life interactions. In the experiment, two sets of movies (AB and BC pairs) featuring a shared character were used to investigate the temporal dynamics of forming integrated and segregated memory representations and their relationship with subsequent memory accuracy. Behaviorally, they found that source memory performance was positively correlated with AC memory inference performance. Using time-resolved representational similarity analysis (RSA), they found that the neural representations of AB and BC movies share both similarities and dissimilarities. During the encoding of the novel character C, there were neural similarities to the AB representation, which predicted successful AC memory inference. In contrast, during the presentation of the common character B, neural dissimilarities to the AB representation were observed, and these predicted accurate source memory. Lastly, univariate analyses revealed that changes in alpha-beta power were associated with these neural (dis)similarity patterns. In summary, the authors provide compelling evidence that the brain dynamically and flexibly engages in both memory integration and separation during the encoding of overlapping events. These processes have distinct temporal and electrophysiological signatures and support different, complementary memory functions.

Overall, I find this study is well-motivated and clearly written. The analyses are appropriately conducted and convincing. The findings expand our understanding of the neural mechanisms for forming and maintaining both integrated and segregated memory representations. In particular, temporally separating the neural processing of integration and separation is novel (to my knowledge) and has broad implications for our understanding of the neural mechanisms underlying overlapping memories. I only have a few comments for the authors, which are mostly about the interpretation of the results:

We sincerely appreciate the reviewer's positive comments about our study and manuscript. We are glad that the reviewer found the work to be well-motivated, clearly written, and of potential impact. We carefully considered the reviewer's suggestions regarding the interpretation of the results, and the manuscript has been revised accordingly. These insightful comments have helped us improve both the clarity and the quality of the paper.

1. The neural similarity analysis revealed increased neural similarity between the Sim A&B representation and the Sim C (and Sim C in context) representation. The authors interpret this as evidence of memory integration. I wonder whether it could merely reflect the reactivation of the Sim A&B memory, rather than the formation of an integrated A-B-C representation.

We thank the reviewer for this comment, which allows us to clarify our interpretation of the observed neural similarities between movie AB and Sim C. In this time window, the neural similarity between movie AB and Sim C (as well as Sim C in context) was positively associated with AC inference (at 2.8 s), but negatively associated with source memory (at 3.7 s). We interpret this pattern as reflecting the formation of an integrated ABC memory representation. Our reasoning is based on prior evidence showing that the formation of integrated memory representations facilitates inference but often comes at the cost of reduced source and detail memory (Carpenter & Schacter, 2017, 2018). If the effect were due to the simple reactivation of the AB movie during Sim C, we would expect improved inference *and* preserved or enhanced source memory, as reactivation would provide an additional learning opportunity.

However, this is not what we observed. Instead, the neural similarity was associated with increased inference coupled with reduced source memory.

We have clarified our interpretation in the revised *DISCUSSION* (see page 20, top paragraph continuing from the previous page).

2. The main analyses focused on the neural similarity between the representation of Sim A&B in context and the BC trials. I am somewhat surprised that there was little to no evidence showing similarity to Sim A or Sim B individually during the Sim C segment. More importantly, there appears to be no Sim B-specific similarity during the Sim B and Sim B&C in context segments, where character B was present on screen. This makes the interpretation of neural similarity ambiguous, as the analysis appears unable to capture common visual inputs.

We are grateful that the reviewer raised this important observation. We agree that the RSA approach should be sensitive to shared visual input, and we conducted additional analysis to explicitly confirm this.

To validate the method, we used the AB movies, taking the advantage of their repeated presentations to examine neural similarity under identical visual input. Specifically, we extracted feature vectors for each AB movie segment across repetitions and correlated them with each timepoint of the entire epoch of the same AB movie (from other repetitions) as well as with different AB movies. This analysis used a time-resolved approach consistent with our main RSA analyses (see *METHOD* in the main text for details).

As shown in **Figure R1**, the results confirm that our method captured common visual input in two complementary ways: (1) within-segment similarity: Neural similarity was higher across repetitions of the same segment than across different AB movies, reflecting consistent responses to identical visual input; and (2) generalization driven by visual overlap: ‘Sim A’ showed similarity to ‘Sim A in Context’ and to ‘Sim A& B in Context’, ‘Sim A in Context’ showed similarity to ‘Sim A&B in Context’, and ‘Sim B’ showed similarity to the ‘Sim A in B in Context’.

Together, these findings validate that the RSA approach here employed is indeed sensitive to shared visual input. These additional results have been incorporated into **Supplementary Note 3**.

Figure R1. Neural representational similarity differences between repetitions of the same AB movie and different AB movies. Shaded areas indicate time windows with significant neural pattern similarity.

Importantly, when participants encode the BC movie, neural responses are likely influenced by two potentially opposing cognitive processes. First, memory reactivation/integration: the presentation of overlapping Sim B may trigger reactivation of the previously seen Sim B in the AB movie, producing increased neural similarity. Second, interference control/separation: to avoid confusion with the prior AB movie, participants may actively differentiate the current BC movie from the previous AB movie, producing increased neural dissimilarity. Critically, we do not know the exact time points during the movie at which one process dominates over the other. If these processes occur with roughly equal frequency across trials, their effects could cancel each other out at the group level, providing a plausible explanation for the absence of a significant overall similarity effect, even when visual input is shared between the two movies.

3. Related to the previous comment, I wonder what the neural dissimilarity truly reflects. The authors argue that this dissimilarity is evidence of the segregation of AB and BC memories. If that is the case, why would this dissimilarity appear in the Sim B segment instead of the Sim B&C in context segment, where the full BC memory event is arguably most salient?

We thank the reviewer for this insightful comment. We did not have prior predictions about the specific timing of neural similarities or dissimilarities. Interestingly, our data revealed stronger dissimilarities when the shared element between the AB and BC movies, that is Sim B, was presented on screen. Previous studies have shown that mnemonic interference increases with the degree of overlap between experiences (Ritvo et al., 2019). As a result, memories tend to be represented more distinctly to resolve such interference (Bakker et al., 2008; Kirwan & Stark, 2007). Since Sim B is the same segment appearing in both movie AB and movie BC, it is reasonable to assume that this is the point at which mnemonic interference is strongest. Consequently, memories may be represented more separately at this moment in order to mitigate the interference.

Supporting this interpretation, our univariate analyses revealed increased alpha-beta power during the timing of the dissimilarity for BC compared with XY movies. Increases in alpha-beta power have been consistently related with cognitive control mechanisms underlying interference resolution (Waldhauser et al., 2012). The convergence of RSA and univariate findings therefore strengthens the view that the Sim B segment specifically elicits mnemonic interference and corresponding interference resolution, leading to the observed increase in neural dissimilarity.

These points are now more explicitly elaborated in the revised discussion (see page 21, top paragraph continuing from the previous page).

4. The results show two significant similarity clusters corresponding to the Sim C and Sim C in context segments. Do they reflect the same underlying neural process? More generally, what is the functional difference between the character segment and the character-in-context segment? After the initial learning, can the character segment (e.g., Sim C) alone trigger memory integration, or is the context an essential component for forming overlapping memories?

We thank the reviewer for this thoughtful comment. It is indeed plausible that characters presented in isolation and characters presented within a narrative context evoke distinct neural representations. Movie segments in which characters interact within vivid contexts are likely encoded in a context-specific manner, capturing the broader event structure and supporting memory for the entire episode. In contrast, segments where characters are presented alone and in a static format may elicit more abstract representations, reflecting a generalized agent who

can appear across multiple interactions. This interpretation is consistent with prior literature suggesting that memory representations derived from naturalistic experiences tend to integrate spatiotemporal and relational context (Baldassano et al., 2017, 2018).

In our data, neural similarity to ‘Sim A&B in Context’ increased during both the ‘Sim C’ and ‘Sim C in Context’ segments. However, only the similarity during ‘Sim C in Context’ significantly predicted AC inference performance. This distinction may simply reflect the higher salience of animated, context-rich stimuli compared with static character images, potentially yielding a stronger signal-to-noise ratio in the similarity analysis.

Thus, while the distinction between naturalistic and simplified stimuli appears to have important consequences for memory representations, future studies will be necessary to clarify how this factor influences the balance between memory integration and separation. These considerations have been incorporated into the revised discussion (see page 21, top paragraph continuing from the previous page).

5. The current experimental design nicely separates the memory integration and separation processes in time and examines their relationship to behavior. To bridge the results to the existing literature, it might be worthwhile to discuss the possible mechanism if the BC pair were presented simultaneously (i.e., only a Sim B and C in context segment). Would integration and separation still both occur? If so, would they have different temporal dynamics within that single segment? Or might the two processes occur simultaneously but be supported by different neural circuits?

We thank the reviewer for this thoughtful comment. Indeed, our current design allowed us to temporally separate memory integration and separation processes. Nevertheless, we speculate that the simultaneous presentation of the BC pair (i.e., only a ‘Sim B and C in context’ segment) could still trigger both integration and separation, albeit with temporal patterns that differ from those observed in the present study.

Upon encountering the shared B, integration would be triggered by the reactivation of the prior AB memory, which would facilitate the formation of an integrated ABC representation. This process appears to depend primarily on hippocampal-neocortical interactions that support binding across episodes (Shohamy & Wagner, 2008; Zeithamova et al., 2012). Concurrently, separation processes may be engaged to resolve interference between overlapping AB and BC representations, recruiting cognitive control mechanisms involving prefrontal regions (Ritvo et al., 2019; Wimber et al., 2015). Thus, with simultaneous BC presentation, integration and separation processes likely operate in parallel within the same time window, as they rely on distinct neural substrates. As a result, dissociating these processes under this condition is challenging, which may partly explain why previous studies have typically reported either integration or separation (Brunec et al., 2020; Holmes et al., 2022; Ritvo et al., 2019).

By presenting event components separately, our design allowed us to disentangle these processes temporally and observe their engagement during the encoding of overlapping events. Our data showed that the engagement of integration and separation processes varies with the perceptual similarity between novel information and prior experience. To bridge our findings with existing literature, we have now incorporated this discussion into the manuscript (see page 22, paragraph 1).

6. Given that both integration and separation processes are observed, I wonder if there is any evidence of a trade-off between them. For example, could stronger integration of a BC

memory into an AB memory lead to a weaker separation process in the Sim B segment, which might, in turn, affect source memory performance? Conversely, could stronger integration create a greater need for separation, thereby increasing the observed neural dissimilarity? While this is an interesting question, I would understand if the authors find it to be outside the scope of the current paper.

This is an interesting question. To address it, we conducted a partial correlation analysis at the trial level, controlling for repetition effects (see **Table R1**). These results indicate that neural pattern similarities tend to co-occur within the same trial. However, the relationship between similarity and dissimilarity remains unclear. While the idea of a potential trade-off between integration and separation is compelling, a definitive conclusion would require further investigation. This analysis has been incorporated into **Supplementary Note 6**.

Table R1 Partial correlation coefficients of the similarities and the dissimilarity

	Similarity at 1.0s	Similarity at 2.8s	Similarity at 3.7s
Similarity at 2.8s	0.001		
Similarity at 3.7s	0.038*	0.046**	
Dissimilarity at 7.4s	0.021	-0.016	0.019

* - $p < .05$; ** - $p < .01$

7. The RSA results showed that the neural similarity and dissimilarity measurements predict AC inference and source memory performance. Are these neural markers also associated with memory for episodic details? For example, might a strongly integrated representation come at the cost of memory for details like the character's clothing? Again, I understand if the authors consider this question beyond the scope of the current paper.

We explored whether participants' memory for clothing was related to the neural similarity between the AB and BC movies (see **Supplementary Note 8**). The results revealed that the similarity between AB movie and the 'Sim A in Context' was negatively associated with memory for clothing. However, this effect was not predictive of AC inference performance. Therefore, we do not have evidence to conclude whether this similarity reflects the formation of an integrated representation or whether there is a trade-off between integration and the retention of episodic details. Addressing these questions will require further research.

Reviewer #3 (Remarks to the Author):

Summary

This study makes a timely and valuable contribution to the field of memory research by investigating how the brain encodes episodic events to support multiple mnemonic functions. Leveraging an ABC associative inference paradigm—a framework well-suited to studying memory generalization and inference—the authors employed naturalistic stimuli (simulated real-life movies) and multivariate analysis techniques to explore the flexible encoding of related experiences. Participants first watched a series of short movies depicting interactions between two characters (AB pairs), followed by a second set of scenes involving one familiar and one novel character (BC pairs). This design enabled the testing of indirect (AC) associations, as well as source memory—i.e., whether participants remembered if two characters had ever appeared together. This dual focus allowed the authors to probe two distinct but complementary memory processes: integration, where overlapping events are linked into unified representations, and separation, where distinct experiences are preserved independently.

The use of time-resolved representational similarity analysis (RSA) on EEG data allowed the authors to track the neural dynamics underlying memory formation with high temporal precision. Importantly, the results revealed that neural pattern similarity between AB and BC episodes predicted successful inference of indirect (AC) associations—indicating integration. In contrast, neural dissimilarity predicted accurate source memory, pointing to the preservation of separate memory traces. This double dissociation is both conceptually and methodologically interesting. It suggests that the brain does not rigidly encode experiences in one way but instead adapts flexibly, balancing integration and separation depending on the demands of future retrieval. These findings align with and extend theoretical frameworks suggesting that memory systems must support both generalization (e.g., for inferential reasoning) and specificity (e.g., for accurate source recollection).

The use of naturalistic stimuli adds ecological validity, enhancing the relevance of the findings for real-world memory processes. Moreover, the combination of behavioral and neural data—analyzed with state-of-the-art multivariate methods—demonstrates a sophisticated experimental approach that provides insights not accessible with traditional univariate or trial-averaged analyses. While the presented findings add to a growing body of evidence with potential implications for memory disorders, I have some major and a few minor issues that would need to be addressed and clarified before publishing. I detail these concerns below.

We thank the reviewer for the positive and constructive remarks about our study and paper. We are glad that the reviewer recognizes the conceptual and methodological contributions of our work, including the use of time-resolved RSA to track neural dynamics underlying memory formation, the double dissociation between neural similarity and dissimilarity in supporting integration versus separation, and the ecological validity provided by naturalistic stimuli. We also appreciate the acknowledgment of our combination of behavioral and neural data analyzed with advanced multivariate methods, which provides insights not accessible with traditional univariate approaches. We address the specific concerns raised by the reviewer in detail below.

Major concerns:

1. My primary concern lies with the introduction, which needs to be substantially rewritten to better frame the study's rationale. As it stands, the introduction heavily emphasizes prior fMRI findings, particularly the role of hippocampal subregions in ABC inference paradigms. While this context is somewhat relevant, it reads more like a retrospective interpretation of the current results than a focused theoretical or empirical motivation for the hypotheses tested in this EEG study. Critically, the introduction fails to reference existing M/EEG research that has examined inference and generalization using similar paradigms, including studies employing representational similarity analysis (RSA). These omissions weaken the justification for using EEG and limit the reader's understanding of how this work builds on or ex-tends prior findings especially concerning brain oscillations.

We appreciate the reviewer's feedback on the introduction. In the revised manuscript, the introduction was substantially rewritten to better align with the goals of the current study. Rather than retrospectively interpreting our results through the lens of prior fMRI findings, the revised framing emphasizes the theoretical motivation to study how memory integration and separation unfold during encoding—processes that the EEG method is well-suited to investigate, especially given its sensitivity to temporal dynamics and shifts.

To strengthen the empirical context, we have incorporated relevant M/EEG studies that have examined inference and generalization using paradigms related to ABC inference, including those applying representational similarity analysis (Backus et al., 2016; Heinbockel et al., 2022; Herweg et al., 2020; Jafarpour et al., 2014; Nicolás et al., 2021). These changes clarify how our work builds on and extend prior EEG research on memory integration (see page 3-5).

We are open to including any additional literature that the reviewer or the editor considers relevant to the study.

2. Furthermore, although the authors perform both univariate and multivariate time-frequency analyses, the introduction lacks any predictions of relevant frequency bands—such as theta—which are known to play a key role in memory integration and retrieval. Without grounding the study in this literature, the introduction does not adequately set up the relevance of frequency-specific dynamics to the research questions. A revised introduction should clearly articulate the motivation for using EEG, specify what temporal or spectral in-sights it can offer beyond fMRI, and situate the work within the existing electrophysiological literature on memory generalization and inference.

We thank the reviewer for this suggestion. We agree that the original introduction lacked sufficient grounding in the electrophysiological literature, particularly regarding frequency-specific dynamics relevant to memory inference and integration. In the revised manuscript, we now clearly articulate why EEG is well-suited to address our research questions—specifically, its ability to capture the temporal evolution and spectral content of neural signals during encoding.

Additionally, we now present specific hypotheses about the roles of theta (4-7 Hz) and alpha-beta (8-30 Hz) activity in memory processes. Drawing on prior EEG studies, we discuss how theta activity has been linked to mnemonic integration and associative binding (e.g., during overlapping episode encoding), while alpha-beta activity has been implicated in both memory reinstatement and inhibitory control mechanisms that may support memory separation. This has been added to the text (see page 5).

These revisions strengthen the theoretical motivation for our time-frequency analyses and situate our work within a growing body of M/EEG research on memory generalization, inference, and representational dynamics. We believe these changes provide a clearer rationale for our analytic approach and the contribution of EEG beyond prior fMRI findings.

Generally, the results seem incomplete and demand deeper analyses:

3. A major concern relates to the spatial characterization of the reported neural similarity effects. While I appreciate that the RSA was conducted at the whole-brain level, the results as presented lack any indication of where on the scalp these effects are most prominent. Especially given the introduction's emphasis on specific brain regions and memory-related processing, I would have expected at least a basic breakdown of whether the observed effects are stronger over frontal, parietal, or other regions. To enhance interpretability, I recommend the authors either (1) report the topographical distribution of the similarity effects or (2) re-run the RSA restricted to functionally meaningful electrode clusters (e.g., frontal, central, parietal). This would provide valuable insight into the potential cognitive and neural sources of the effects and better connect the findings to prior literature on memory integration and segregation.

We thank the reviewer for inquiring about the spatial characterization of the reported neural similarity effects. To address this concern, we re-ran the RSA at the channel level. Specifically, we estimated the RSA for each electrode along with its immediate neighbors (for a similar procedure, see Bramão et al., 2025; Bramão & Johansson, 2018) to reconstruct the topographical distribution of neural pattern similarity to 'Sim A&B in Context' during BC movie viewing (see **Figure R2**).

The results showed that both similarity and dissimilarity effects were broadly distributed across the scalp without a clear focal topography. Notably, this widespread topography aligns with the univariate analysis results (see **Figure 6**), suggesting that while these processes may originate from distinct hippocampal subregions, they likely engage widespread cortical networks and reflect large-scale neural activity.

This topographical analysis has been added to the supplementary materials (see **Supplementary Note 4**) to provide additional insight into the spatial characteristics of the observed effects.

Figure R2 Topographical distribution of similarity and dissimilarity between 'Sim A&B in Context' and BC movie.

4. Closely related to the previous point, it is unclear why no source reconstruction analyses were conducted, especially given that the experiment was designed around RSA with a clear theoretical focus on specific brain regions, notably the hippocampus. While I understand

that the RSA was performed at the sensor level, this approach limits anatomical interpretability—particularly in a study aiming to link EEG pattern similarity to memory mechanisms commonly associated with deep brain structures. Given the relevance of medial temporal regions in memory integration and inference, I strongly recommend that the authors complement their current analyses with source-level RSA or at least conduct source localization of the key effects. This would significantly strengthen the anatomical specificity of the findings and align the EEG results more directly with the fMRI literature that motivated the study.

We appreciate the reviewer's suggestion to include source reconstruction analyses to enhance the anatomical interpretation of our findings. We fully agree that linking EEG-based RSA results to underlying brain structures is an important goal, especially given the central role of medial temporal regions in memory integration and inference.

However, we ultimately decided not to pursue source-level RSA in this study for both methodological and empirical reasons. First, as the reviewer notes, integration and separation are processes likely triggered by the hippocampus—a deep brain structure that is impossible to investigate with scalp EEG.

Second, our sensor-level analyses revealed broadly distributed effects across the scalp, without clear focal topographies that would guide targeted source reconstruction. In the absence of strong, a priori constraints, or distinct topographical patterns, we were concerned that source localization would introduce uncertainty without yielding interpretable anatomical specificity.

That said, we agree this is an important direction for future research. In particular, combining EEG with complementary imaging modalities (e.g., fMRI-constrained source modelling or intracranial recordings) could help clarify the neural generators of the observed RSA effects. We acknowledge these limitations and future directions in the Discussion section (see page 23).

5. Given the well-known limitations of detecting hippocampal activity with EEG—even when employing advanced source reconstruction methods such as LCMV beamformers—I believe it would be valuable for the authors to explore functional connectivity of relevant cortical regions, such as the angular gyrus, in relation to the reported effects. Incorporating connectivity analyses would enhance the interpretive depth of the findings and provide a more comprehensive picture of the neural mechanisms underlying the observed behavioral outcomes.

We thank the reviewer for this thoughtful suggestion. Following the recommendation, we explored whether functional connectivity analyses could provide additional insight into the neural mechanisms underlying representational similarity and dissimilarity.

We conducted a dynamic functional connectivity analysis at the channel level. Specifically, we estimated connectivity between all channel pairs at each time point by correlating their decibel transformed power values across the 3–35 Hz frequency range, using Pearson correlation followed by fisher-z transformation for statistical purposes. This yielded a time course of dynamic functional connectivity for each trial throughout the duration of BC movie. XY movie was also investigated and included as a baseline.

To evaluate condition effects, we first computed the difference in connectivity between BC and XY encoding, averaged within each segment, using paired t-test with Bonferroni corrections to p -values (see **Figure R3**); however, no significant difference emerged.

Next, to assess whether connectivity dynamics were predictive of representational similarity, we concatenated the trial-wise connectivity time courses during BC encoding for each

participant and used them to predict the neural pattern similarity between AB and BC movies (concatenated in the same way). Using the Bayesian linear regression (with Normal-Inverse-Gamma distribution prior, $\alpha = 15$, $\beta = 15$, $\mu = 0$, $\lambda = I * 30$, I is unit matrix), we calculated the regression coefficient matrix reflecting the contribution of each connectivity pair in predicting similarity values (see **Figure R4**). However, no functional connectivity was predictive of neural pattern similarity.

Taken together, the connectivity analyses did not provide conclusive evidence for either condition-related differences or a systematic relationship to neural similarity/dissimilarity. For this reason, we have opted not to include them in the paper. However, if the reviewer or editor believes that these analyses would still add value, we would be glad to incorporate them as supplementary material.

Figure R3 Channel-wise dynamic functional connectivity within each segment during BC/XY encoding.

Figure R4. Standardized Bayesian Regression results showing how dynamic functional connectivity predicts neural pattern similarity.

Minor concerns:

1. It would be helpful if the authors addressed potential effects of the repeated presentation (5×) of the stimuli. Since neural signals were averaged across repetitions, the role of novelty vs. familiarity—a key factor in memory encoding—should be considered. The neural dynamics of the first versus the fifth presentation may differ substantially, and this could impact both the behavioral and neural findings. A brief analysis or at least a discussion of how repetition may have influenced RSA results are needed.

We thank the reviewer for raising this point. Repeated exposure to the movie clips likely strengthened event representations and could have modulated the neural dynamics underlying memory formation. To examine this, we analyzed the trajectory of neural pattern similarity and dissimilarity across the five repetitions (**Supplementary Note 5**).

The results indicated that changes were not linear across all repetitions. Instead, neural representations showed incremental updating during the first few viewings, after which they stabilized. This pattern could be consistent with novelty-driven updating processes being most prominent early on, while later repetitions reflected more stable, familiarity-based retrieval.

We have now incorporated this consideration into the Discussion (see page 21, top paragraph continuing from the previous page), highlighting how novelty versus familiarity may have influenced the RSA results.

2. The behavioral data in Figure 3 appear to exhibit substantial ceiling effects, particularly in accuracy scores. The authors should clarify whether the use of linear mixed models (LMMs) is appropriate given the limited variance. Please elaborate on how this was handled in the analysis and interpretation.

We acknowledge that ceiling effects were present in the behavioral data, but they were confined to the direct association trials, particularly the AB condition. Crucially, the primary focus of the study lies in the indirect AC association trials and source memory, where no ceiling effects were observed.

Regarding the use of linear mixed models (LMMs), we conducted model-fitting tests to evaluate the contribution of each factor to model performance and to ensure that potential confounds were controlled (see page 26, *METHOD* → *Behavioral Data Analysis and Statistics*, paragraph 1, and **Supplementary Note 2**). See also the minor comment 1 from Reviewer 1. These steps mitigate concerns that the limited variance in the direct association trials could bias the overall conclusions. Moreover, since our key findings are based on the indirect AC associations, where performance showed sufficient variability, we believe the use of LMMs remains appropriate for the current data.

3. Response times for BC retrieval were notably slower than for AB. This difference deserves further discussion. Could this be due to fatigue, re-encoding demands, or greater cognitive load during BC trials? Including a plausible interpretation would help contextualize this finding.

We thank the reviewer for highlighting this effect. As noted, participants' response times were significantly longer for BC trials compared to both AB and XY trials.

Because BC and XY movies were presented in an interleaved fashion during encoding, it is unlikely that the slower BC responses reflect fatigue or general task disengagement. A more plausible explanation is that BC trials elicited stronger interference due to the overlapping Sim

B, which likely reactivated competing AB associations. This retrieval competition could have increased retrieval demands, consistent with prior work showing that proactive interference and conflict resolution can prolong response times in associative memory tasks (Bramão et al., 2025; Liu et al., 2024). We have now incorporated this discussion into the main text (see page 9, *Results* → *Behavioral Results* → *Associative Memory Test*, paragraph 2).

4. In the caption of Figure 2, there is a minor typo:

“(D) Systematic similarities, indicative of memory integration should predict of AC retrieval performance.” This should be corrected to:“(D) Systematic similarities, indicative of memory integration, should be predictive of AC retrieval performance.”

The text was revised accordingly.

5. Did the authors assess the perceptual or narrative similarity between the SIM movies and stories (AB, BC pairs)? If some stimuli were more similar than others, this could confound RSA results. It would strengthen the study to report whether similarity ratings (e.g., by independent raters) were obtained and whether these were controlled for or included in the analyses.

We appreciate the reviewer’s thoughtful question regarding potential perceptual or narrative similarity between the Sim movies and how this might influence RSA results. While we did not obtain independent similarity ratings for the AB, BC, or XY movies, we took care to address this concern through our experimental design.

Specifically, the assignment of AB/BC and XY movies was fully counterbalanced across participants, such that each movie pair could appear in different conditions (AB/BC or XY) across individuals. This design ensured that any inherent perceptual or narrative similarity between specific movies was not confounded with the condition (i.e., BC vs. XY). In other words, across the full sample, each movie pair was equally likely to appear as AB/BC or XY condition, minimizing the impact of stimulus-specific similarity on the RSA results.

This is now clearly explained in the text (see page 25, paragraph 1).

References

- Ambrosius, W. T. (Ed.). (2007). *Topics in biostatistics*. Humana Press.
- Audrain, S., & McAndrews, M. P. (2022). Schemas provide a scaffold for neocortical integration of new memories over time. *Nature Communications*, *13*(1), 5795. <https://doi.org/10.1038/s41467-022-33517-0>
- Backus, A. R., Schoffelen, J.-M., Szebényi, S., Hanslmayr, S., & Doeller, C. F. (2016). Hippocampal-Prefrontal Theta Oscillations Support Memory Integration. *Current Biology*, *26*(4), 450–457. <https://doi.org/10.1016/j.cub.2015.12.048>
- Bakker, A., Kirwan, C. B., Miller, M., & Stark, C. E. L. (2008). Pattern Separation in the Human Hippocampal CA3 and Dentate Gyrus. *Science*, *319*(5870), 1640–1642. <https://doi.org/10.1126/science.1152882>
- Baldassano, C., Chen, J., Zadbood, A., Pillow, J. W., Hasson, U., & Norman, K. A. (2017). Discovering Event Structure in Continuous Narrative Perception and Memory. *Neuron*, *95*(3), 709–721.e5. <https://doi.org/10.1016/j.neuron.2017.06.041>
- Baldassano, C., Hasson, U., & Norman, K. A. (2018). Representation of Real-World Event Schemas during Narrative Perception. *The Journal of Neuroscience*, *38*(45), 9689–9699. <https://doi.org/10.1523/JNEUROSCI.0251-18.2018>
- Bramão, I., Jiang, J., Wagner, A. D., & Johansson, M. (2022). Encoding contexts are incidentally reinstated during competitive retrieval and track the temporal dynamics of memory interference. *Cerebral Cortex*, *32*(22), 5020–5035. <https://doi.org/10.1093/cercor/bhab529>
- Bramão, I., & Johansson, M. (2018). Neural Pattern Classification Tracks Transfer-Appropriate Processing in Episodic Memory. *ENEURO*, *5*(4), ENEURO.0251-18.2018. <https://doi.org/10.1523/ENEURO.0251-18.2018>
- Bramão, I., Liu, Z., & Johansson, M. (2025). Remembering the past affects new learning: The temporal dynamics of integrative encoding. *Neuropsychologia*, *212*, 109148. <https://doi.org/10.1016/j.neuropsychologia.2025.109148>
- Brunec, I. K., Robin, J., Olsen, R. K., Moscovitch, M., & Barense, M. D. (2020). Integration and differentiation of hippocampal memory traces. *Neuroscience & Biobehavioral Reviews*, *118*, 196–208. <https://doi.org/10.1016/j.neubiorev.2020.07.024>
- Carpenter, A. C., & Schacter, D. L. (2017). Flexible retrieval: When true inferences produce false memories. *Journal of Experimental Psychology: Learning, Memory, and Cognition*, *43*(3), 335–349. <https://doi.org/10.1037/xlm0000340>
- Carpenter, A. C., & Schacter, D. L. (2018). False memories, false preferences: Flexible retrieval mechanisms supporting successful inference bias novel decisions. *Journal of Experimental Psychology: General*, *147*(7), 988–1004. <https://doi.org/10.1037/xge0000391>
- Grandchamp, R., & Delorme, A. (2011). Single-Trial Normalization for Event-Related Spectral Decomposition Reduces Sensitivity to Noisy Trials. *Frontiers in Psychology*, *2*. <https://doi.org/10.3389/fpsyg.2011.00236>
- Heinbockel, H., W.E.M. Quaedflieg, C., Wacker, J., & Schwabe, L. (2022). Spatio-temporal theta pattern dissimilarity in the right centro-parietal area during memory generalization. *Brain and Cognition*, *164*, 105926. <https://doi.org/10.1016/j.bandc.2022.105926>
- Herweg, N. A., Solomon, E. A., & Kahana, M. J. (2020). Theta Oscillations in Human Memory. *Trends in Cognitive Sciences*, *24*(3), 208–227. <https://doi.org/10.1016/j.tics.2019.12.006>
- Holmes, N. M., Wong, F. S., Bouchekioua, Y., & Westbrook, R. F. (2022). Not “either-or” but “which-when”: A review of the evidence for integration in sensory

- preconditioning. *Neuroscience & Biobehavioral Reviews*, *132*, 1197–1204.
<https://doi.org/10.1016/j.neubiorev.2021.10.032>
- Hu, L., Xiao, P., Zhang, Z. G., Mouraux, A., & Iannetti, G. D. (2014). Single-trial time–frequency analysis of electrocortical signals: Baseline correction and beyond. *NeuroImage*, *84*, 876–887. <https://doi.org/10.1016/j.neuroimage.2013.09.055>
- Jafarpour, A., Fuentemilla, L., Horner, A. J., Penny, W., & Duzel, E. (2014). Replay of Very Early Encoding Representations during Recollection. *The Journal of Neuroscience*, *34*(1), 242–248. <https://doi.org/10.1523/JNEUROSCI.1865-13.2014>
- Kirwan, C. B., & Stark, C. E. L. (2007). Overcoming interference: An fMRI investigation of pattern separation in the medial temporal lobe. *Learning & Memory*, *14*(9), 625–633. <https://doi.org/10.1101/lm.663507>
- Kuhl, B. A., Rissman, J., Chun, M. M., & Wagner, A. D. (2011). Fidelity of neural reactivation reveals competition between memories. *Proceedings of the National Academy of Sciences*, *108*(14), 5903–5908. <https://doi.org/10.1073/pnas.1016939108>
- Lachaux, J.-P., Lutz, A., Rudrauf, D., Cosmelli, D., Le Van Quyen, M., Martinerie, J., & Varela, F. (2002). Estimating the time-course of coherence between single-trial brain signals: An introduction to wavelet coherence. *Neurophysiologie Clinique/Clinical Neurophysiology*, *32*(3), 157–174. [https://doi.org/10.1016/S0987-7053\(02\)00301-5](https://doi.org/10.1016/S0987-7053(02)00301-5)
- Lee, C. S., Aly, M., & Baldassano, C. (2021). Anticipation of temporally structured events in the brain. *eLife*, *10*(e64972).
- Liu, Z., Johansson, M., Johansson, R., & Bramão, I. (2024). The effects of episodic context on memory integration. *Scientific Reports*, *14*(1), 30159. <https://doi.org/10.1038/s41598-024-82004-7>
- Matuschek, H., Kliegl, R., Vasishth, S., Baayen, H., & Bates, D. (2017). Balancing Type I error and power in linear mixed models. *Journal of Memory and Language*, *94*, 305–315. <https://doi.org/10.1016/j.jml.2017.01.001>
- Nicolás, B., Sala-Padró, J., Cucurell, D., Santurino, M., Falip, M., & Fuentemilla, L. (2021). Theta rhythm supports hippocampus-dependent integrative encoding in schematic/semantic memory networks. *NeuroImage*, *226*, 117558. <https://doi.org/10.1016/j.neuroimage.2020.117558>
- Pacheco Estefan, D., Sánchez-Fibla, M., Duff, A., Principe, A., Rocamora, R., Zhang, H., Axmacher, N., & Verschure, P. F. M. J. (2019). Coordinated representational reinstatement in the human hippocampus and lateral temporal cortex during episodic memory retrieval. *Nature Communications*, *10*(1), 2255. <https://doi.org/10.1038/s41467-019-09569-0>
- Ritvo, V. J. H., Turk-Browne, N. B., & Norman, K. A. (2019). Nonmonotonic plasticity: How memory retrieval drives learning. *Trends in Cognitive Sciences*, *23*(9), 726–742. <https://doi.org/10.1016/j.tics.2019.06.007>
- Shohamy, D., & Wagner, A. D. (2008). Integrating memories in the human brain: Hippocampal-midbrain encoding of overlapping events. *Neuron*, *60*(2), 378–389. <https://doi.org/10.1016/j.neuron.2008.09.023>
- Waldhauser, G. T., Johansson, M., & Hanslmayr, S. (2012). Alpha/Beta Oscillations Indicate Inhibition of Interfering Visual Memories. *The Journal of Neuroscience*, *32*(6), 1953–1961. <https://doi.org/10.1523/JNEUROSCI.4201-11.2012>
- Wimber, M., Alink, A., Charest, I., Kriegeskorte, N., & Anderson, M. C. (2015). Retrieval induces adaptive forgetting of competing memories via cortical pattern suppression. *Nature Neuroscience*, *18*(4), 582–589. <https://doi.org/10.1038/nn.3973>
- Zeithamova, D., Dominick, A. L., & Preston, A. R. (2012). Hippocampal and Ventral Medial Prefrontal Activation during Retrieval-Mediated Learning Supports Novel Inference. *Neuron*, *75*(1), 168–179. <https://doi.org/10.1016/j.neuron.2012.05.010>